

# Local forcing mechanisms challenge parameterizations of ocean thermal forcing for Greenland tidewater glaciers

Alexander O. Hager[1], David A. Sutherland[1], and Donald A. Slater[2]

[1]Department of Earth Sciences, University of Oregon, OR, USA
[2]School of Geosciences, University of Edinburgh, Edinburgh, UK

**Correspondence:** ahager@uoregon.edu

**Abstract.** Frontal ablation has caused $32 - 66\%$ of Greenland Ice Sheet mass loss since 1972, and despite its importance in driving terminus change, ocean thermal forcing remains crudely incorporated into large-scale ice sheet models. In Greenland, local fjord-scale processes modify the magnitude of thermal forcing at the ice-ocean boundary but are too small scale to be resolved in current global climate models. For example, simulations used in the Ice Sheet Intercomparison Project for CMIP6 (ISMIP6) to predict future ice sheet change rely on the extrapolation of regional ocean water properties into fjords to drive terminus ablation. However, the accuracy of this approach has not previously been tested due to the scarcity of observations in Greenland fjords, as well as the inability of fjord-scale models to realistically incorporate icebergs. By employing the recently developed IceBerg package within the MITgcm, we here evaluate the ability of ocean thermal forcing parameterizations to predict thermal forcing at tidewater glacier termini. This is accomplished through sensitivity experiments using a set of idealized Greenland fjords each forced with equivalent ocean boundary conditions, but with varying tidal amplitudes, subglacial discharge, iceberg coverage, and bathymetry. Our results indicate that the bathymetric obstruction of external water is the primary control on near-glacier thermal forcing, followed by iceberg submarine melting. We find that grounding line thermal forcing varies by $2.9°C$ across all simulations and is heavily dependent on the depth of bathymetric sills in relation to the Polar-Atlantic Water thermocline. However, using a common adjustment for fjord bathymetry we can still predict grounding line thermal forcing within $0.2°C$ in our simulations. Finally, we introduce new parameterizations that account for iceberg-driven cooling that can accurately predict interior fjord thermal forcing profiles both in iceberg-laden simulations and in observations from Ilulissat Icefjord.

## 1 Introduction

Mass loss from the Greenland Ice Sheet (GrIS) contributed $10.8 \pm 0.9$ mm to mean sea level rise from 1992 to 2018 (The IMBIE Team, 2019) and is projected to raise sea level by $90 - 180$ mm by 2100 (Fox-Kemper et al., 2021). This mass loss has, in part, been triggered by the tidewater glacier response to warming ocean temperatures (e.g., Nick et al., 2009; Holland et al., 2008; Murray et al., 2010; Motyka et al., 2011; Straneo and Heimbach, 2013; Wood et al., 2018), with frontal ablation accounting for $32 - 66\%$ of GrIS mass loss since 1972 (Enderlin et al., 2014; Van den Broeke et al., 2016; Mouginot et al., 2019). In Greenland, fjords are the principal pathways connecting tidewater glacier termini to the coastal ocean, in which local



processes relating to sill-driven mixing and silled obstruction of external water (Mortensen et al., 2011, 2013, 2014; Moffat
      et al., 2018; Jakobsson et al., 2020; Hager et al., 2022), submarine melting of icebergs and glacier termini (Davison et al., 2020;
      Jackson et al., 2020; Magorrian and Wells, 2016; Moon et al., 2018), and subglacial discharge (Carroll et al., 2015; Jenkins,
      2011) modulate the magnitude of ocean forcing at the ice-ocean boundary, often on a seasonal basis (e.g., Moffat et al., 2018;
      Mortensen et al., 2013; Hager et al., 2022). However, such processes are too small scale to be resolved in global climate

models (e.g., Watanabe et al., 2010; Golaz et al., 2019), and instead, sea level rise projections have relied on poorly-validated
      simplifying parameterizations of oceanic boundary conditions in ice sheet models (e.g., Morlighem et al., 2019; Jourdain et al.,
      2020) that create large sources of uncertainty when predicting future mean sea levels (Goelzer et al., 2020; Seroussi et al.,
      2020). This paper focuses on the ocean thermal forcing of GrIS outlet glaciers, yet Antarctic ice sheet models face similar
      challenges when prescribing ocean boundary conditions beneath ice shelves (e.g., Seroussi et al., 2020; Jourdain et al., 2020;

Burgard et al., 2022).

      Recent studies have shown that multi-decadal retreat across a population of tidewater glaciers can be reasonably approxi-
      mated as a linear function of the climate forcing they experience (Cowton et al., 2018; Slater et al., 2019; Fahrner et al., 2021;
      Black and Joughin, 2022). For many Greenland tidewater glaciers, change in terminus position is specifically thought to be
      the result of enhanced submarine melting of the terminus and subsequent changes to ice dynamics (e.g., Holland et al., 2008;

Murray et al., 2010; Straneo and Heimbach, 2013; Luckman et al., 2015; Smith et al., 2020). Taken together, these findings
      have prompted the development of parameterizations that use submarine melting to drive frontal ablation in ice sheet models.
      In particular, the Ice Sheet Intercomparison Project for CMIP6 (ISMIP6) (Nowicki et al., 2016, 2020), which produced sea
      level contribution projections for Greenland in the 6th Assessment Report of the Intergovernmental Panel on Climate Change
      (Masson-Delmotte et al., 2021), relies on two such parameterizations. The first parameterization (called the retreat implemen-

tation) is the simplest – being designed to be implementable in all participating ISMIP6 models – and is used to determine
      changes in glacier terminus position over a given time (Slater et al., 2019, 2020):

$$\Delta L = \kappa \left( Q_{t_2}^{0.4} \, \Theta_{t_2} - Q_{t_1}^{0.4} \, \Theta_{t_1} \right) \tag{1}$$

      where $\Delta L$ is the retreat/advance distance (km) between times $t_1$ and $t_2$, $Q$ is the mean summer subglacial discharge, $\Theta$ is the
      ocean thermal forcing (°C above freezing temperature), and $\kappa$ is a coefficient tuned to fit the observed terminus positions of
      almost 200 Greenland tidewater glaciers between 1960–2018 (Slater et al., 2019). Here, submarine melting is assumed to scale

proportionally to $Q^{0.4}\Theta$.

      The second parameterization, deemed the submarine melt implementation, encompasses only submarine melt, and leaves
      the subsequent glacier response (as given by the relationship between ice flux, submarine melt, and calving) to be calculated
      by the ice sheet model (e.g., Morlighem et al., 2016). Here, submarine melt rate ($\dot{m}$) is (Rignot et al., 2016):

$$\dot{m} = \left( 3 \times 10^{-4} \, h \, q^{0.39} + 0.15 \right) \Theta^{1.18} \tag{2}$$

      where $h$ is grounding line depth and $q$ is the annual mean subglacial discharge normalized by calving front area. In both

implementations, ice sheet models need a method for prescribing $\Theta$ based on offshore ocean conditions. In ISMIP6, this was



done by first taking a spatial average of annual mean ocean conditions within seven large regional zones surrounding Greenland (Slater et al., 2020). For the retreat implementation (Eq. 1), glaciers are forced with a depth-averaged $\Theta$ so that all glaciers within a region experience the same thermal forcing. In the submarine melt implementation (Eq. 2), an adjustment is made accounting for fjord bathymetry preventing deep currents from reaching the glacier face (Section 2.2; Morlighem et al., 2019).

However, neither parameterization explicitly incorporates water transformation between the coast and glacier termini (e.g., Gladish et al., 2015; Straneo et al., 2012), which can vary greatly even between neighboring fjords (Bartholomaus et al., 2016). Furthermore, accelerated mass loss from the Greenland Ice Sheet can be largely attributed to the dynamics of only a small number of individual glaciers (Enderlin et al., 2014; Fahrner et al., 2021), which can dominate uncertainty of regional retreat projections (Goelzer et al., 2020). There is thus an urgent need for improved parameterizations that incorporate local water

transformation and that are validated by high-resolution models or extensive observations.

The large-scale and long-term observations necessary to validate such parameterizations are logistically difficult in Greenland, suggesting a modeling approach is warranted. However, until recently, general circulation models lacked the ability to realistically incorporate iceberg melting, which can be the primary freshwater source in Greenland fjords (Enderlin et al., 2016; Moon et al., 2018). Here, we employ the newly developed IceBerg package (Davison et al., 2020) within the Massachusetts

Institute of Technology general circulation model (MITgcm) (Marshall et al., 1997) that enables the inclusion of icebergs to test the accuracy of both ISMIP6 thermal forcing parameterizations across a variety of local forcing conditions. We create a suite of idealized model simulations each forced with different combinations of subglacial discharge, iceberg prevalence, tidal forcing, and sill geometry, but all experiencing the same offshore temperature and salinity conditions at the open boundary. In doing so, our objective is to simulate the diverse array of neighboring fjord conditions that can result from the same regional

ocean forcing when local factors are accounted for. We then quantify the error of each ISMIP6 thermal forcing parameterization for all model runs and determine the primary contributors to local water transformation and uncertainty within each formulation. Based on our results, we recommend simple improvements to current thermal forcing parameterizations that substantially improve their accuracy.

## 2 Methods

### 2.1 Model Setup

MITgcm fjord geometries were typical of Greenland fjords (e.g., Straneo and Cenedese, 2015) and were 800 m deep, 5 km wide, 60 km long, and had a laterally uniform, Gaussian-shaped sill near the mouth of the fjord with a minimum depth of either 100 m, 250 m, or 400 m (hereafter distinguished as S100, S250, and S400 runs; Figure 1). We did not include runs without sills because preliminary simulations showed no difference from S400 runs. Vertical resolution was 10 m in the upper 100 m,

20 m between 100 – 500 m depth, and 50 m below 500 m. The majority of runs had horizontal resolutions of 200 m; however, a few high subglacial discharge and iceberg meltwater flux runs were conducted at 500 m resolution to avoid running at an impractical timestep (Table C1). An 800 m deep coastal zone was constructed outside the fjord with an additional 30 cells to the west and 20 cells to the north and south. Horizontal resolution in the coastal zone linearly telescoped to 2 km at the



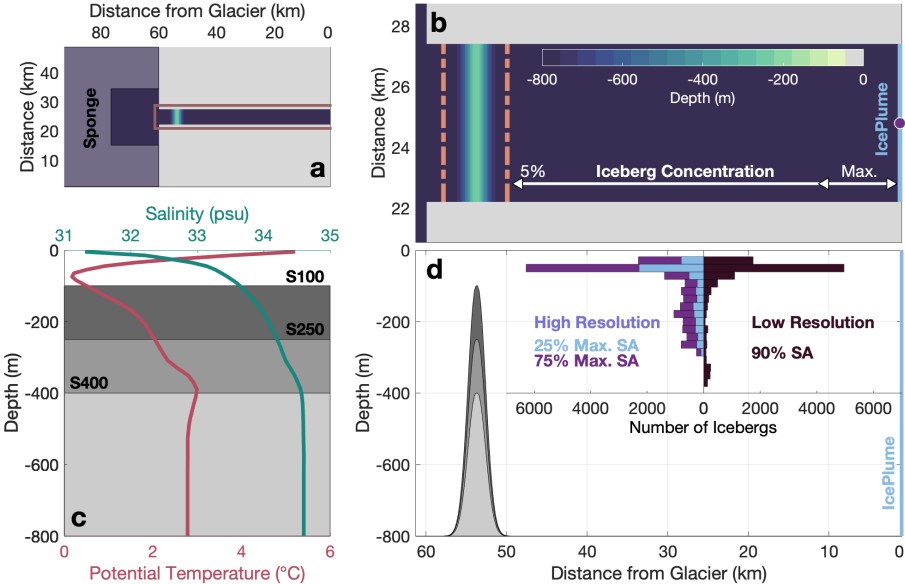

**Figure 1. (a)** Model domain with the location of the sponge layer along open boundaries. **(b)** Enlargement of brown box in **a** depicting the locations of the entrance sill, Total Exchange Flow transects (dashed lines), glacier face (blue line), subglacial discharge plume center-point (purple dot), and the distribution of iceberg concentrations. **(c)** Initial and open boundary conditions in relation to the depths of each sill. **(d)** Along-fjord cross-section of **b** depicting the vertical distribution of iceberg keel depths (binned every 25 m) for each iceberg scenario (labeled by the maximum coverage of grid cell surface area, SA), plotted with the depths of each sill.

northern, western, and southern open boundaries. A 10 cell sponge layer was imposed at each open boundary to inhibit internal

waves from reflecting back into the domain.

All simulations were run in a hydrostatic configuration with a nonlinear free surface and a Coriolis frequency of $1.3752 \times 10^{-4}$ s$^{-1}$. High and low resolution simulations were run at timesteps of $25 - 30$ s and $60$ s, respectively. Horizontal viscosity was prescribed using a Smagorinsky scheme (Smagorinsky, 1963) and a Smagorinsky constant of 2.2, while horizontal diffusivities were set to zero (though numerical diffusion will still exist). We used the KPP parameterization (Large et al., 1994)

for vertical mixing, setting the background and maximum viscosity to $5 \times 10^{-4}$ s$^{-1}$ and $5 \times 10^{-3}$ s$^{-1}$, respectively, and background and maximum diffusivities to zero and $5 \times 10^{-5}$ s$^{-1}$, respectively. Simulations were run until all fjord water below sill depth had been flushed and water properties stopped evolving ($200 - 1000$ days). Output was averaged over the last 10 days of model time, and all "near-glacier" model output was averaged over the two closest grid cells to the glacier face.

Simulations were initialized from temperature and salinity data profiles observed in $2013 - 2015$ outside the Uummannaq

fjord system, West Greenland (Bartholomaus et al., 2016), which shares a similar vertical structure to summer coastal properties around Greenland: a warm, fresh summer surface layer underlain by cold Polar Water and warm, salty Atlantic Water at depth (Figure 1c; Straneo et al., 2012; Straneo and Cenedese, 2015). The same profiles were used as boundary conditions along the open boundaries (Figure 1a). M2 frequency tidal velocities of $5 \times 10^{-3}$ m s$^{-1}$ were imposed along the western boundary,



creating tidal amplitudes of $\sim 1.5$ m within the fjord, typical of tides throughout East and West Greenland (Howard and
Padman, 2021). For fjord geometries where significant tidal mixing was expected (S100 and S250 simulations), we tested
additional high and low tidal forcing scenarios with tidal velocities of $7 \times 10^{-3}$ m s$^{-1}$ and $3 \times 10^{-3}$ m s$^{-1}$, creating tides of
2.06 m and 0.88 m, respectively (Table S1).

Subglacial discharge and glacier submarine melting were parameterized with the IcePlume package (Cowton et al., 2015)
using a straight glacier face along the eastern boundary (Figure 1b, d). Summer high resolution runs were forced with subglacial
discharge of 300 m$^3$ s$^{-1}$, which is typical of summer values from Kangilliup Sermia (Rink Isbræ) (Bartholomaus et al., 2016;
Carroll et al., 2016). Summer low resolution runs were designed to resemble the largest Greenland glacial fjords and were
forced with subglacial discharge of 1000 m$^3$ s$^{-1}$, characteristic of glacier runoff entering Sermilik Fjord and Ilulissat Icefjord
(Echelmeyer and Harrison, 1990; Gladish et al., 2015; Carroll et al., 2016; Moon et al., 2018). Subglacial discharge plumes are
parameterized to have a half-conical geometry in most runs; however, we test the influence of plume geometry (and thus near-
terminus subglacial hydrology) by repeating five runs with subglacial discharge spread out across a 1 km wide line-plume (e.g.,
Jenkins, 2011), which may be more realistic for some fjord systems (Jackson et al., 2017; Hager et al., 2022; Kajanto et al.,
2022). Winter scenarios were reinitialized from the steady state of summer runs with the same tidal, iceberg, and geometric
constraints, but were forced by a 10 m$^3$ s$^{-1}$ line-plume across the entire glacier width to account for a switch to basal friction-
generated, distributed subglacial drainage in the winter (e.g., Cook et al., 2020). To be consistent with ISMIP6 methodology,
we do not account for seasonal differences in offshore waters; thus, our seasonal sensitivity runs only test seasonal variation in
subglacial discharge. In all runs, a background velocity of 0.1 m s$^{-1}$ was implemented to facilitate ambient submarine melting
along the glacier face.

Icebergs were parameterized using the IceBerg package (Davison et al., 2020), which treats icebergs as stationary barriers
to flow and adjusts surrounding fjord water properties according to calculated meltwater fluxes. Iceberg depths were set using
an inverse power law size frequency distribution with an exponent of -1.9, similar to those observed in Kangilliup Sermia
and Sermilik fjords (Sulak et al., 2017). Consistent with observations (e.g., Enderlin et al., 2016; Sulak et al., 2017; Moon
et al., 2018; Schild et al., 2021), we prescribed a maximum iceberg depth of 300 m and 400 m in high and low resolution runs,
respectively (Figure 1d). Icebergs were concentrated at the fjord head, filling either 25% or 75% of the fjord surface area within
10 km of the glacier, which linearly decreased to 5% just inside the entrance sill (Figure 1c). These concentrations approximate
those observed at Kangilliup Sermia and Sermilik fjords (Sulak et al., 2017). Additional S250 simulations targeting the ice-
choked conditions of Ilulissat Icefjord were conducted at low resolution using an iceberg concentration of 90% throughout the
fjord. Meltwater plumes resulting from iceberg submarine melt were parameterized by imposing a background velocity of 0.06
m s$^{-1}$ along the iceberg face. All forcing and geometric conditions were repeated with and without icebergs. See Table C1 for
a full list of all 27 model simulations.

**2.2 Testing of ISMIP6 Thermal Forcing Parameterizations**

For comparison to our simulations, we calculate the thermal forcing that would have been imposed in ISMIP6 experiments
assuming regional water properties equal to our open boundary conditions. In the first thermal forcing parameterization (IS-



MIP6retreat; Table 1) used with Eq. 1, thermal forcing is determined by:

$$\Theta(z) = \theta(z) - \theta_f(z) = \theta(z) - [\lambda_1 S(z) + \lambda_2 + \lambda_3 z] \tag{3}$$

where $\theta$ and $S$ are the prescribed potential temperature and practical salinity profiles at the open boundaries, $\theta_f$ is the local
freezing temperature at depth $z$, and $\lambda_1 = -5.73 \times 10^{-2}\,°\text{C psu}^{-1}$, $\lambda_2 = 8.32 \times 10^{-2}\,°\text{C}$, and $\lambda_3 = 7.61 \times 10^{-4}\,°\text{C m}^{-1}$ (Jenkins,
2011). Profiles of $\Theta(z)$ are then depth-averaged between 200 – 500 m depth (Slater et al., 2020) to provide a singular value,
$\Theta_{\bar{z}} = 4.7°\text{C}$ (Table 1), across all simulations for ISMIP6retreat. This range of depth was chosen to encompass most Greenland
tidewater glacier grounding lines (Slater et al., 2019).

In contrast to ISMIP6retreat, the submarine melt thermal forcing parameterization (ISMIP6melt) accounts for bathymetry
preventing external water from entering the fjord below sill depth (Table 1; Morlighem et al., 2019). This is accomplished by
first defining an effective depth as the deepest part of the near-glacier water column in direct contact with the open ocean (here
equal to the sill depth). Fjord water properties above the effective depth are directly extrapolated to the glacier terminus, while
water properties below sill depth are made equal to those at the effective depth (e.g., Morlighem et al., 2019). Extrapolated
potential temperature and practical salinity are then converted to *in-situ* temperature and absolute salinity before calculating
thermal forcing across the glacier face: $\Theta(z) = T(z) - T_f(z)$. Here, $T$ and $T_f$ are the *in-situ* temperature and freezing temper-
ature, which together with the absolute salinity are calculated using the non-linear TEOS-10 toolbox (McDougall and Barker,
2011). As in Slater et al. (2020), the final $\Theta$ value used in Eq. 2 is taken from the grounding line depth. The ISMIP6melt
formulation therefore predicts the same thermal forcing for all runs within each of the S100 (2.9°C), S250 (4.6°C), and S400
(5.5°C) groups (Figure 2). We compare the ISMIP6 thermal forcings with a number of quantities extracted from our simu-
lations; these are: (1) $\Theta_{\overline{gl}}$, modeled near-glacier thermal forcing at the grounding line; (2) $\Theta_{\bar{z}}$, modeled near-glacier thermal
forcing averaged between 200 – 500 m depth; and (3) $\Theta_{\overline{A}}$, modeled area-mean, near-glacier thermal forcing (Table 1).

### 2.3    Quantification of Sill-driven Mixing

Following MacCready et al. (2021), Hager et al. (2022), and Bao and Moffat (2023) we quantify the net effect of sill-driven
vertical mixing by pairing the estuarine Total Exchange Flow framework (TEF) (MacCready, 2011) with efflux/reflux theory
(Cokelet and Stewart, 1985). We use this approach because it provides bulk mixing transports that are easily relatable to
other forcing processes. TEF utilizes isohaline coordinates to identify inflowing and outflowing transports that satisfy the
Knudsen Relations and account for both tidal and subtidal fluxes (Knudsen, 1900; MacCready, 2011; Burchard et al., 2018).
We use 1000 salinity classes to bin salt and volume fluxes across each transect, and employ the dividing salinity method
(MacCready et al., 2018; Lorenz et al., 2019) to calculate inward and outward transports, allowing for the potential for multiple
layers of each to exist. Inflowing and outflowing transport-weighted salinities are given by $S_{in,out} = F^s_{in,out}/Q_{in,out}$, where
$F^s_{in,out}$ and $Q_{in,out}$ are the sums of all inflowing and outflowing salt and volume fluxes, respectively. We treat temperature as
a tracer corresponding to each salt class so that the transport-weighted inflowing and outflowing temperatures are calculated
as: $T_{in,out} = F^t_{in,out}/Q_{in,out}$ (e.g., Lorenz et al., 2019).



**Table 1.** Descriptions of ISMIP6 thermal forcing parameterizations, new thermal forcing parameterizations presented in this paper, and thermal forcing metrics extracted from our simulations.

**ISMIP6 Parameterizations**

| Name | Description |
|---|---|
| ISMIP6retreat | Boundary conditions averaged between $200 - 500$ m depth. |
| ISMIP6melt | Boundary conditions above sill extrapolated to glacier face. Near-glacier water properties below sill are made equal to boundary conditions at sill depth. Thermal forcing is defined at the grounding line. |

**New Parameterizations**

| Name | Description |
|---|---|
| AMretreat | Area-mean boundary conditions across the glacier face. |
| AMmelt | Same as ISMIP6melt, but thermal forcing is defined as an area-mean across the glacier face. |
| AMberg | Same as AMmelt, but temperatures in the upper 175 m are adjusted to follow the Gade slope before averaging. |
| AMconst | Same as AMmelt, but temperatures the upper 175 m are set equal to the temperature at 175 m depth before averaging. |
| AMfit | AMberg used where icebergs are prevalent and AMmelt used where icebergs are scarce. |

**Thermal Forcing Metrics**

| Name | Description |
|---|---|
| $\Theta_{\overline{gl}}$ | Modeled near-glacier thermal forcing at the grounding line. |
| $\Theta_{\overline{z}}$ | Modeled near-glacier thermal forcing averaged between $200 - 500$ m depth. |
| $\Theta_{\overline{A}}$ | Modeled area-mean, near-glacier thermal forcing. |

Efflux/reflux theory assumes an estuarine system where mixing is concentrated at constrictions (such as sills) separated by deep basins (dubbed reaches) where mixing is minimal (Cokelet and Stewart, 1985). At each mixing zone, some portion of





inflowing or outflowing water is vertically mixed and recirculated, or refluxed, into the opposing layer and back into its original

reach, while the remainder, the efflux, is transported across the mixing zone to the next reach (Figure A1). Using mass and

volume conservation, the percentage of inflowing or outflowing water that is refluxed or effluxed can be written in terms of TEF

variables (MacCready et al., 2021), but for the purposes of this paper, we are primarily concerned with $\alpha_{out}^r$, which represents

the percent of the outflowing fjord water that is refluxed at the entrance sill:

$$\alpha_{out}^r = \frac{Q_{in}^g}{Q_{out}^g} \frac{S_{in}^o - S_{in}^g}{S_{in}^o - S_{out}^g} \tag{4}$$

where superscripts $o$ and $g$ denote the TEF transports on the oceanward and glacierward sides of the mixing zone, respectively

(Figure A1). We calculated efflux/reflux budgets between two TEF transects on either side of the entrance sill, and avoid pre-

scribing icebergs within the sill region to ensure temperature and salt are conserved across the mixing zone. More information

about using TEF with efflux/reflux theory can be found in MacCready et al. (2021), Hager et al. (2022), and in Appendix A.

## 2.4   Calculation of Local Heat Fluxes

Quantifying the heat fluxes associated with each local forcing mechanism is important when determining the primary causes

of local water transformation. The heat flux resulting from the submarine melting of ice ($H_{melt}$) is calculated by:

$$H_{melt} = -\rho_{mw} Q_{mw}[L + c_i(\theta_f - \theta_i)] \tag{5}$$

where $\rho_{mw}$ is the meltwater density (1000 kg m$^{-3}$), $Q_{mw}$ is the total meltwater flux as determined from IceBerg or IcePlume,

$L$ is the latent heat of fusion, $c_i$ is the heat capacity of ice, $\theta_f$ is the potential freezing temperature, and $\theta_i$ is the potential

temperature of ice. In our experiments, ice is set to its melting temperature so that Eq. 5 collapses to

$$H_{berg} = -\rho_{mw} Q_{berg} L \tag{6}$$

and

$$H_{sm} = -\rho_{mw} Q_{sm} L \tag{7}$$

for the iceberg and glacier submarine melt heat fluxes, respectively.

Advective heat fluxes ($H_{adv}$) arising from sill-driven reflux and subglacial discharge are given by:

$$H_{adv} = \rho c_p Q (\theta_{adv} - \theta_r) \tag{8}$$

where $\rho$ is the water density, $c_p$ is the heat capacity of water, $Q$ is the advective volume flux, $\theta_{adv}$ is the potential temperature

of the advected fluid, and $\theta_r$ is a reference temperature. To calculate the recirculatory heat flux caused by sill-driven mixing

(here called heat reflux), we substitute TEF quantities into Eq. 8 so that:

$$H_{reflux} = \rho c_p \alpha_{out}^r Q_{out}^g (T_{out}^g - T_{in}^g) \tag{9}$$



where $\alpha_{out}^r Q_{out}^g$ is the reflux volume, $T_{out}^g - T_{in}^g$ is the temperature difference between the outflowing and inflowing layers on the sill's glacierward side, and $\rho$ is the refluxed water density as determined from $T_{out}^g$, $S_{out}^g$, and the mid-column water pressure at 400 m depth. Here, $H_{reflux}$ refers to the heat transfer (positive or negative) from the outflowing layers to inflowing layers on the sill's glacierward side as a result of sill-driven mixing. For the subglacial discharge heat flux, Eq. 8 is specified as:

$$H_{sg} = \rho_{sg} \, c_p \, Q_{sg} \, (T_{sg} - T_{in}^g) \tag{10}$$

where $Q_{sg}$ is the total subglacial discharge, $\rho_{sg}$ is 1000 kg m$^{-3}$, $T_{sg}$ is 0°C, and the reference temperature, $T_{in}^g$, is chosen to be consistent with Eq. 9. In practice, $T_{in}^g$ works well as a reference temperature in both Eqs. 9 and 10 as inflowing water properties remain largely unaltered between the sill and glacier face.

### 2.5 Empirical Orthogonal Functions of Near-Glacier Variability

Empirical Orthogonal Function (EOF) analysis was conducted on near-glacier thermal forcing profiles to determine the dominant modes of variability between runs. Near-glacier $\Theta(z)$ profiles were horizontally averaged across the calving face before removing the mean $\Theta$ within each sill group (Figure 3a). This second step was necessary to account for the dependence of mean fjord temperatures on sill depth (Figure 2). EOFs were then calculated on the resultant profiles.

## 3 Results

### 3.1 Near-Glacier Water Properties

Despite identical temperature and salinity forcing at the open boundaries, thermal forcing averaged over the glacier face ($\Theta_{\bar{A}}$; Table 1) varied by 2.7°C across all runs (Figure 2; Table C1). Grounding line thermal forcing ($\Theta_{\overline{gl}}$; Table 1) varied by 2.9°C between runs, while grounding line salinities differed by 1.4 psu (Figure 2). In S400 runs, near-glacier water properties were largely unmodified from the open boundary conditions, particularly below 200 m where the influence of subglacial discharge and iceberg melt was negligible. However, water properties below sill depth progressively freshened and cooled as the sill depth shoaled, allowing for $\Theta_{\bar{A}}$ to be neatly grouped by depth of sill: $4.2 - 4.7$°C for S400 runs, $3.6 - 4.1$°C for S250 runs, and $2.0 - 2.9$°C for S100 runs (Figure 2).

Water properties below sill depth were nearly homogeneous in all runs, with only minor variability occurring when iceberg keels extended below sill depth or subglacial discharge plumes reached neutral buoyancy below sill depth (this most often occurs with line-plumes). The greatest variability existed above 175 m, where most iceberg melting occurred and where most summer subglacial discharge plumes reached neutral buoyancy (Figure 2).

Unsurprisingly, iceberg runs were always cooler than their non-iceberg counterparts; however, the difference between these groups was larger for runs with shallower sills. On average, the difference in $\Theta_{\bar{A}}$ between iceberg and non-iceberg runs diminished from 0.7°C for S100 runs to 0.3°C and 0.2°C for S250 and S400 runs, respectively. Iceberg melt had the greatest impact on water properties in the upper 175 m, contributing to a temperature range of 5.1°C at the surface, independent of sill



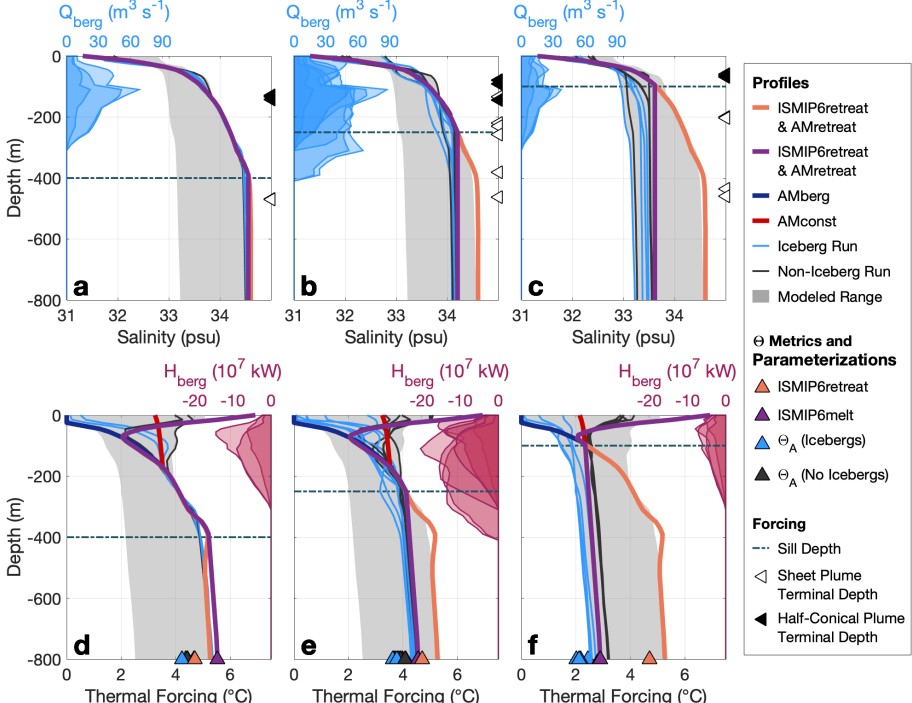

**Figure 2.** Near-glacier **(a-c)** salinity and **(d-f)** thermal forcing profiles for all iceberg (light blue) and non-iceberg (gray) runs, plotted with the profiles used to calculate ISMIP6retreat/AMretreat (orange) and ISMIP6melt/AMmelt (purple). The orange profile is also equivalent to the boundary conditions. In **d-f**, the profiles used to calculate AMberg and AMconst (Section 4.2) are depicted in dark blue and red, respectively. The gray background in all plots illustrates the range across all runs and the horizontal dashed line depicts sill depth (**a,d** are S400 runs; **b,e** are S250 runs; **c,f** are S100 runs). Triangles in **a-c** represent the terminal plume depth for line-plumes (white) and half-conical plumes (black). Orange and purple triangles in **d-f** depict $\Theta$ for ISMIP6retreat and ISMIP6melt, respectively, in relation to $\Theta_{\bar{A}}$ for iceberg (blue triangles) and non-iceberg (gray triangles) runs. The vertical distribution of iceberg freshwater fluxes ($Q_{berg}$) and heat fluxes ($H_{berg}$) are shown in **a-c** and **d-f**, respectively.

.

depth. However, where iceberg keel depth exceeded sill depth, iceberg melting cooled the entire water column to the grounding line, indicating some volume of iceberg meltwater was mixed and refluxed at the silled mixing zone. Such cooling is most

pronounced in S100 runs, where the difference in grounding line thermal forcing was on average 0.5°C between iceberg and non-iceberg runs. $\Theta_{\bar{A}}$ showed no significant difference between tidal sensitivity runs, neither was there an appreciable distinction between runs with sheet and half-conical plumes and otherwise equivalent forcing. Winter runs were generally cooler than their summer counterparts in the upper water column, with $\Theta_{\bar{A}}$ varying by $\leq 0.3$°C between winter and summer discharge scenarios with otherwise equivalent forcing (Table C1).

After removing the dominant influence of sills (Figure 3a), EOF analysis indicates the presence or absence of icebergs accounts for 84% of the remaining near-glacier thermal forcing variability between runs (Figure 3b). In general, this first EOF



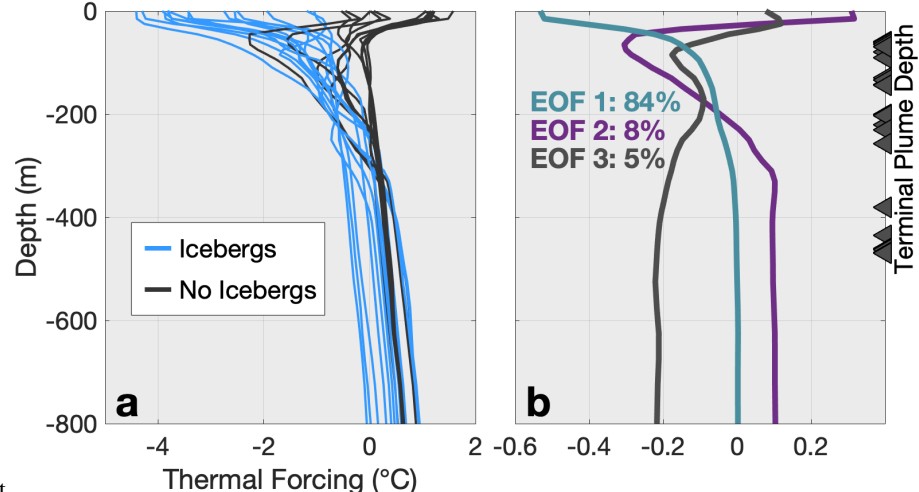

t

**Figure 3. (a)** Near-glacier thermal forcing profiles from all iceberg (light blue) and non-iceberg (gray) runs after removing the depth-average of each sill group. **(b)** The three dominant EOF modes with the percentage of variance they contribute, as calculated from the profiles in **a**. Gray triangles indicate terminal plume depths of all runs. The teal line represents variance from iceberg melting, the purple line indicates variance stemming from the boundary conditions, and the gray line signifies variance from subglacial discharge plumes.

mode reflects the same pattern of cooling in the upper 300 m present in all iceberg runs, but its amplitude changes sign for non-iceberg runs, thus imitating the warm surface water that exists when icebergs are absent (Figure 3). The second EOF mode makes up $8\%$ of the variance between runs and has a spatial structure identical to the open boundary temperature conditions

(Figure 3b). Therefore, the second mode can be interpreted as the influence of regional ocean temperatures on near-glacier thermal forcing, in large part accounting for the minimal fjord water mass transformation in S400 non-iceberg runs. A third EOF mode contributing $5\%$ of the variance depicts temperature variability coincident with the terminal depths of subglacial discharge plumes (Figure 3b), and is therefore interpreted to represent variable outflowing plume conditions. As water below sill depth is homogeneous, variability resulting from reflux is difficult to discern through EOF analysis. It is therefore possible

that reflux variability is incorporated into any of the three dominant modes, part of the remaining $3\%$ of the variance without clear physical corollaries, or was removed with the mean during EOF computation.

## 3.2 Internal Freshwater Sources and Heat Fluxes

Subglacial discharge and iceberg meltwater had similar contributions ($\sim 30 - 65\%$) to the total freshwater input into summer iceberg runs, with glacial meltwater contributing less than $4\%$. In contrast, iceberg melt flux was the dominant freshwater

source ($\sim 95\%$ of all freshwater fluxes) in winter iceberg runs. In non-iceberg runs, subglacial discharge made up $\geq 90\%$ of freshwater input in the summer, while glacier submarine melt was the dominant freshwater source in the winter ($53 - 65\%$ of all freshwater fluxes).



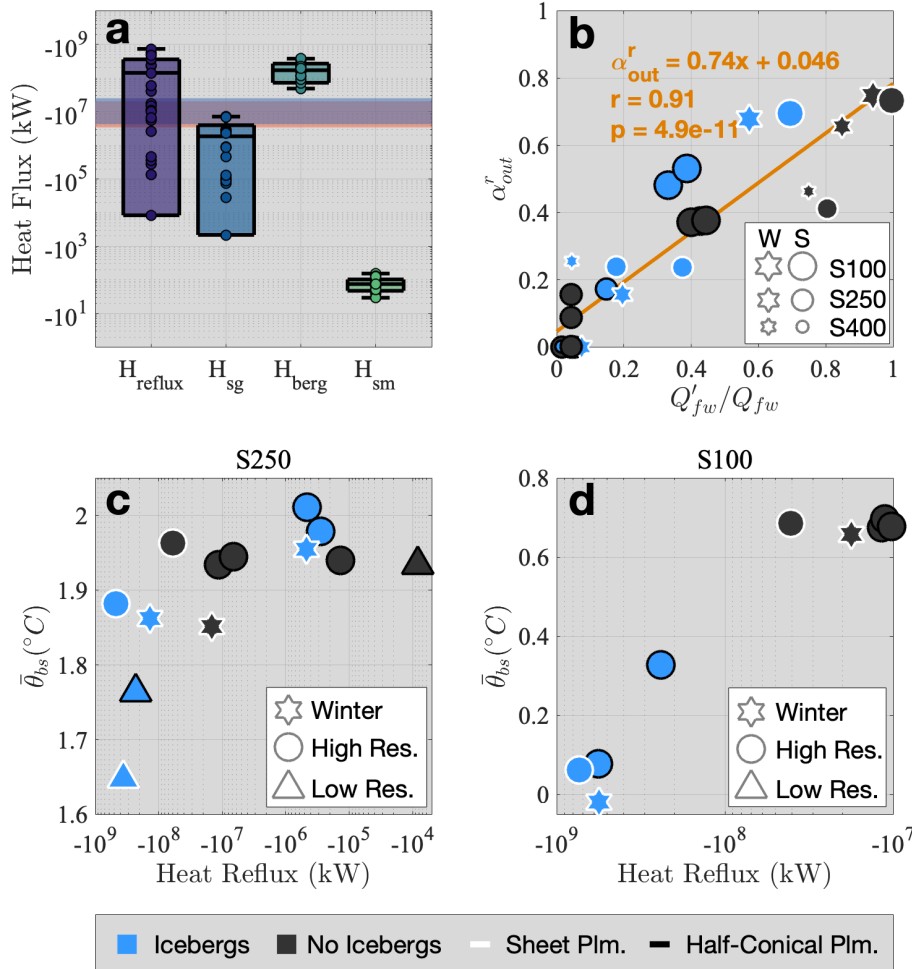

**Figure 4.** **(a)** Box plots of heat fluxes associated with reflux ($H_{reflux}$), subglacial discharge ($H_{sg}$), iceberg submarine melting ($H_{berg}$), and glacier submarine melting ($H_{sm}$). Box plots depict maximum, minimum, mean, and standard deviations across all runs. Horizontal blue and orange rectangles depicts ranges of estimated surface heat fluxes (Section 4.1) for winter and summer runs, respectively. Note that summer surface heat fluxes are reflected across the x-axis for illustrative purposes, but are actually positive and represent surface warming. **(b)** Reflux fraction ($\alpha_{out}^r$) as a function of $Q'_{fw}/Q_{fw}$, the portion of freshwater input released below sill depth. Stars and circles differentiate between winter and summer runs, respectively. Marker sizes vary by sill depth. Orange line is Eq. 11. **(c-d)** Depth-averaged, near-glacier potential temperatures below sill depth ($\bar{\theta}_{bs}$) as a function of heat reflux for S250 and S100 runs. Marker shape differentiates between winter (stars), summer high resolution (circles), and summer low resolution (triangles) runs. In **b-d**, light blue and gray markers represent iceberg and non-iceberg runs, respectively, and white edges depict runs forced with a line-plume.

Despite comparable freshwater fluxes in summer runs, the heat flux from iceberg submarine melting surpassed that from subglacial discharge by multiple orders of magnitude, regardless of iceberg concentration or subglacial discharge (Figure 4a). Iceberg melting removed heat from surrounding waters at rates of $-4.9 \times 10^7$ to $-3.8 \times 10^8$ kW. In contrast, subglacial



discharge heat fluxes were $-2.2 \times 10^3$ to $-7.2 \times 10^6$ kW and glacier submarine melt heat fluxes were -29 to -150 kW. Heat reflux spanned five orders of magnitude from $-8.3 \times 10^3$ to $-7.4 \times 10^8$ kW, at its maximum exceeding the magnitudes of even the largest iceberg heat fluxes, while at its minimum falling near the lower limits of subglacial discharge heat flux.

As opposed to heat fluxes from freshwater sources, which principally cool the upper water column, heat reflux can directly
facilitate cooling of deep water. Our experiments show a pronounced cooling of deep water temperatures with increasingly negative heat reflux for both S250 and S100 runs, resulting in a decrease of over $\sim 0.3°$C and $\sim 0.6°$C, respectively (Figure 4c–d). In general, the runs with the highest heat reflux either contained icebergs, line-plumes, or both; however, there is no clear relationship between heat reflux and any specific local forcing processes (Figure 4a–b). Nevertheless, there is a highly significant ($r = 0.91$, $p = 4.9 \times 10^{-11}$) linear relationship between the portion of freshwater input released below sill depth
and the percent of outflowing water refluxed at the entrance sill, $\alpha_{out}^r$ (Figure 4b). In our experiments, $\alpha_{out}^r$ can be estimated by:

$$\alpha_{out}^r = 0.74 \frac{Q'_{fw}}{Q_{fw}} + 0.046 \tag{11}$$

where $Q_{fw} = Q_{sg} + Q_{sm} + Q_{berg}$ is the total freshwater input and the prime denotes freshwater entering the domain below sill depth (subglacial discharge is included in $Q'_{fw}$ if the plume reaches neutral buoyancy below sill depth).
All S100 runs had significant sill-driven reflux ($\alpha_{out}^r \geq 37\%$). $\alpha_{out}^r$ in S250 runs ranges from $0 - 66\%$, but is highest in runs with substantial iceberg freshwater flux below sill depth, or in runs with sheet-like plumes (Figure 4b). $\alpha_{out}^r$ was negligible in all summer S400 runs, but became significant in winter runs where weak subglacial discharge plumes still intersected the sill at depth (Table C1). Despite equivalent $\alpha_{out}^r$ between S100 tidal sensitivity runs ($\sim 37\%$), tidal forcing does have a minor effect on S250 runs and is responsible for a range of $0.2 - 16\%$ in $\alpha_{out}^r$ across S250 tidal scenarios (Table C1).

### 3.3   Thermal Forcing Parameterizations

Overall, ISMIP6melt accurately predicted $\Theta_{\overline{gl}}$ within a root mean square error (RMSE) of $0.2°$C, an error that varied by $\pm 0.08°$C regardless of sill depth or iceberg prevalence (Table 2). ISMIP6retreat predicted mean near-glacier thermal forcing between $200 - 500$ m depth ($\Theta_{\bar{z}}$) within an RMSE of $1.35°$C (Table 2). This error was minimal (RMSE $= 0.10°$C) for S400 runs, in which fjord water was of similar composition to shelf water, but increased substantially with successively shallower
sills (RMSE $= 0.71°$C for S250 runs and RMSE $= 2.17°$C for S100 runs; Table 2).

Both parameterizations are designed to at least crudely target thermal forcing at glacier grounding lines for two reasons: (1) buoyant upwelling along the glacier terminus may homogenize the near-glacier water column (e.g., Mankoff et al., 2016), and (2) grounding line thermal forcing may promote undercutting, and thus calving, of the glacier face. However, the best depth at which to prescribe thermal forcing at a calving face is still a topic of ongoing debate; therefore, we also employ a skill score
(SS) to asses each parameterization's ability to predict thermal forcing throughout the entire water column:

$$SS = 1 - \frac{\frac{1}{N} \Sigma_{i=1}^{i=N} (p_i - m_i)^2}{\frac{1}{N} \Sigma_{i=1}^{i=N} (\mid p_i - \bar{m} \mid + \mid m_i - \bar{m} \mid)^2} \tag{12}$$



where $m_i$ is the MITgcm or observed near-glacier thermal forcing at z-coordinate $i$, $\bar{m}$ is the mean of all $m_i$ values, $p_i$ is the corresponding value predicted by each parameterization, and $N$ is the number of z-coordinates in a profile. An SS value of 1 indicates perfect agreement, while a value of 0 indicates no agreement between profiles. Both profiles used to compute ISMIP6melt and ISMIP6retreat had moderate success at parameterizing the near-glacier thermal forcing profile, with average skill scores of 0.5–0.6 across all runs (Table 2). Skill scores for iceberg runs were relatively poor (SS = 0.44–0.49) compared to non-iceberg runs (SS = 0.57–0.73). Skill scores also steadily decreased with sill depth from $\sim 0.70$ in S400 runs to 0.31–0.46 in S100 runs (Table 2).

**Table 2.** Root mean square error (RMSE) and skill score of thermal forcing parameterizations for different groups of model runs. The RMSE of ISMIP6melt is calculated relative to $\Theta_{\overline{gl}}$, while the RMSE of ISMIP6retreat is calculated relative to $\Theta_{\bar{z}}$. The RMSE of all other parameterizations is relative to $\Theta_{\bar{A}}$.

**Root Mean Square Error**

| Parameterization | Overall | Icebergs | No Icebergs | S400 | S250 | S100 |
|---|---|---|---|---|---|---|
| | (°C) | (°C) | (°C) | (°C) | (°C) | (°C) |
| AMmelt | 0.30 | 0.40 | 0.13 | 0.26 | 0.24 | 0.39 |
| AMretreat | 1.24 | 1.37 | 1.08 | 0.17 | 0.66 | 1.98 |
| AMberg | 0.26 | 0.18 | 0.34 | 0.15 | 0.18 | 0.38 |
| AMconst | 0.31 | 0.38 | 0.21 | 0.29 | 0.27 | 0.36 |
| **AMfit** | **0.15** | **0.18** | **0.13** | **0.10** | **0.09** | **0.23** |
| ISMIP6melt[*] | 0.20 | 0.19 | 0.21 | 0.19 | 0.12 | 0.28 |
| ISMIP6retreat [‡] | 1.35 | 1.42 | 1.26 | 0.10 | 0.71 | 2.17 |

[*] RMSE is calculated relative to $\Theta_{\overline{gl}}$.    [‡] RMSE is calculated relative to $\Theta_{\bar{z}}$.

**Skill Score**

| Parameterization | Overall | Icebergs | No Icebergs | S400 | S250 | S100 |
|---|---|---|---|---|---|---|
| AMmelt | 0.58 | 0.44 | 0.73 | 0.70 | 0.61 | 0.46 |
| AMretreat | 0.53 | 0.49 | 0.57 | 0.69 | 0.62 | 0.31 |
| AMberg | 0.64 | 0.91 | 0.35 | 0.83 | 0.68 | 0.49 |
| AMconst | 0.69 | 0.69 | 0.68 | 0.84 | 0.73 | 0.54 |
| **AMfit** | **0.82** | **0.91** | **0.73** | **0.91** | **0.83** | **0.76** |
| ISMIP6melt | 0.58 | 0.44 | 0.73 | 0.70 | 0.61 | 0.46 |
| ISMIP6retreat | 0.53 | 0.49 | 0.57 | 0.69 | 0.62 | 0.31 |



## 4 Discussion

### 4.1 Uncertainty in Thermal Forcing Parameterizations

Our results demonstrate the wide range of fjord conditions that can result from equivalent regional ocean forcing and emphasize the need for local processes to be incorporated into the coupling of global climate and ice sheet models. In order of importance, we identify these local processes as:

1. *Bathymetric obstruction of external water* – The $2.7°$C range in $\Theta_{\bar{A}}$ in our runs is strongly dependent on the depth of the entrance sill, which preferentially blocks warm water from entering the fjord when the sill is shallow. The prominent thermocline between $100\,\text{m} - 400\,\text{m}$ depth in our boundary conditions (Figure 1c) has been observed in fjords throughout Greenland, and marks the transition between Polar and Atlantic water (Straneo et al., 2012). In our experiments, external temperatures range from $0.46°$C at 100 m to $3°$C at 400 m depth (Figure 1c), nearly the exact range in $\Theta_{\bar{A}}$ between S100 runs and S400. Furthermore, we see no overlap in $\Theta_{\bar{A}}$ between the three sill depths. Therefore, we conclude that the depth of the entrance sill in relation to the $100\,\text{m} - 400\,\text{m}$ thermocline plays a first-order role in determining internal fjord temperatures, but we do not expect sill depth to be a strong control when sills lie below the Polar–Atlantic water thermocline.

2. *Presence or absence of icebergs* – After adjusting for the silled obstruction of external water, cooling from iceberg meltwater (or lack thereof) is responsible for $84\%$ of the remaining variability between runs (Figure 3), as well as a temperature difference of $5.1°$C at the surface. Iceberg-driven cooling primarily occurs in the upper 175 m of the water column; however, where iceberg keels extend below sill depth, cooling affects the entire water column as a result of sill-driven reflux. Similar magnitudes of iceberg-driven cooling were modeled in Davison et al. (2020), Davison et al. (2022), and Kajanto et al. (2022). Iceberg meltwater fluxes are comparable to subglacial discharge in summer runs and dominate the freshwater budget in the winter, which is in agreement with prior estimates of iceberg meltwater fluxes in Greenland fjords (Enderlin et al., 2016; Moon et al., 2018). However, the additional energy required to melt this volume of water enables icebergs to disproportionately cool the surrounding water column when compared to a similar volume flux of subglacial discharge.

3. *Refluxed outflowing water* – Heat reflux can rival the heat flux of iceberg melting, leading to a substantial cooling of deep water temperatures. Where heat reflux is important, the $\lesssim 0.6°$C cooling occurs throughout the water column from sill depth to the grounding line, often affecting a much larger portion of the water column than the melting of icebergs. While such an effect is hard to identify with EOF analysis, the decrease of deep water temperatures with heat reflux (Figure 4c, d) indicates this process has the potential to significantly impact near-glacier thermal forcing in certain fjords.

4. *Subglacial discharge* – While subglacial discharge certainly affects near-glacier thermal forcing, the variability (both inter- and intra-seasonal) this contributes to near-glacier temperature profiles is overshadowed by the influence of ice-



bergs. EOF analysis suggests subglacial discharge is responsible for only $5\%$ of the near-glacier temperature variability. However, subglacial discharge remains a critical driver of submarine melting through its influence on glacial fjord circulation (Carroll et al., 2015; Sciascia et al., 2013; Straneo et al., 2011; Xu et al., 2012), deep water renewal (Carroll et al., 2017), turbulent upwelling (Slater et al., 2015), near-glacier horizontal circulation (Slater et al., 2018), and enhanced iceberg melt (Kajanto et al., 2022). Our results further indicate the terminal depth of subglacial discharge plumes can

directly affect reflux at silled mixing zones (Figure 4b), and thus influence deep water temperatures.

5. *Surface heat flux* – Although we neglect surface heat fluxes in our model, we approximate their magnitude based on previous estimates from Sermilik Fjord. Surface heat fluxes in Sermilik Fjord are thought to be $\sim 80$ W m$^{-2}$ in the summer and $\sim -100$ W m$^{-2}$ in the winter (Jackson and Straneo, 2016; Hasholt et al., 2004). Applying these values to the exposed surface area (not covered by icebergs) in our simulations, we estimate total surface heat fluxes in our

simulations would be $3.4 \times 10^6$ to $2.0 \times 10^7$ kW in the summer and $-4.3 \times 10^6$ to $-2.5 \times 10^7$ kW in the winter (Figure 4a). Thus, surface heat fluxes could often exceed those from subglacial discharge but are an order of magnitude less than iceberg melt heat fluxes. However, surface heating only affects the uppermost water column, and thus we do not expect it to substantially affect $\Theta_{\bar{A}}$ in Greenland fjords. Nevertheless, surface heating may significantly affect heat reflux in Alaska, Svalbard, and Patagonia fjords where shallow sills protrude into the surface layer (e.g., Hager et al., 2022; Bao

and Moffat, 2023).

6. *Glacier submarine melting* – While our modeling suggests glacier submarine melting has little effect on near-glacier thermal forcing, we cannot discount it as an important variable. Our model resolution is too coarse to resolve the complexities of the ice-ocean boundary, and recent observations indicate the IcePlume package may substantially underestimate ambient melting (Jackson et al., 2020).

Observational studies support these model-identified drivers of contrasting water properties between nearby fjords. In northwest Greenland, slight differences in sill geometry allow warm Atlantic Water to flow unimpeded into Petermann Fjord, while inflow of Atlantic Water into the inner basin of neighboring Sherard Osborn Fjord is restricted to a cross-sectional area $\sim 7.5 \times$ smaller (Jakobsson et al., 2020). When paired with enhanced sill-driven reflux of glacially-modified water, this restricted heat inflow is responsible for a $0.2°$C difference in near-glacier water between these otherwise similar fjords (Jakobsson et al.,

2020). Furthermore, Kangilliup Sermia, Kangerlussuup Sermia, and Sermeq Kujalleq (hereafter referring to Jakobshavn Isbræ) are all located within the same ISMIP6 region, and are thus subject to the equivalent ocean thermal forcing in ISMIP6 projections (except for bathymetric adjustments used for ISMIP6melt) (Slater et al., 2020). Yet, mean fjord temperatures in the upper 100 m differed by up to $2.5°$C in summer 2014, seemingly due to the large iceberg meltwater flux into Ilulissat Icefjord, where Sermeq Kujalleq terminates (Bartholomaus et al., 2016; Mojica et al., 2021; Kajanto et al., 2022).

In addition to local forcing processes, our results point to a further source of error in the commonly used thermal forcing parameterizations, namely the dependence on specific depths when calculating thermal forcing based on regional profiles. For ISMIP6retreat, a depth-average of regional ocean thermal forcing from 200–500 m was chosen to encompass most Greenland



tidewater glacier grounding lines (Slater et al., 2020), yet this depth range also bounds the largest vertical temperature gradient along the Greenland coast. Thus, any sill that exists within or above this range will greatly affect the ability of ISMIP6retreat to accurately predict near-glacier thermal forcing. Indeed, ISMIP6retreat was quite accurate for S400 runs, in which the difference in $200 - 500$ m temperatures between external and internal water was minimal, but its accuracy decreased progressively with each shallower sill as the temperature difference between external and internal water increased to $> 1.5°C$.

The current state of frontal ablation laws used in large-scale ice sheet models requires a single scalar to represent thermal forcing across the entire glacier terminus (e.g., Aschwanden et al., 2019; Choi et al., 2021). Unfortunately, this makes it difficult to define the best metric for prescribing near-glacier thermal forcing. Current methods often rely on grounding line conditions, motivated by the fact that grounding line submarine melt may promote undercutting and calving (e.g., Rignot et al., 2016; O'Leary and Christoffersen, 2013); however, glacier termini can actually exhibit a wide range of geometries resulting from spatially variable melt rates (Fried et al., 2019), which can either increase or decrease calving depending on the magnitude and shape of the entire melt profile (Ma and Bassis, 2019). Thus, an overemphasis on undercutting processes may lead to spurious frontal ablation laws. Furthermore, the choice to prioritize grounding line thermal forcing in both ISMIP6melt and ISMIP6retreat negates the iceberg-driven cooling of the upper water column, which our modeling indicates is the secondary control on near-glacier thermal forcing. While buoyant upwelling may partially homogenize near-terminus waters (at a finer scale than our model resolution), particularly within the subglacial discharge plume (e.g., Mankoff et al., 2016), it seems likely that iceberg meltwater remains an important control on thermal forcing at the ice-ocean boundary. Although it remains unclear at what depth glacier dynamics are most sensitive to ocean thermal forcing, from an ocean forcing perspective, it may be prudent to define thermal forcing using a metric that captures all processes relevant to near-glacier conditions, such as iceberg melting. Such a tactic would also pave the way for coupled mélange-ice dynamics models, which would require accurate portrayal of the upper water column to appropriately model mélange dynamics and prescribe back stress to the glacier. Therefore, we here test alternative methods for parameterizing a full profile of ocean thermal forcing, as well as an area-mean, which both allow for the inclusion of all dominant fjord-scale processes, while remaining relevant to calving processes.

## 4.2 Alternative Thermal Forcing Parameterizations

We first test revisions to each ISMIP6melt and ISMIP6retreat that are independent of specific depths, here called Area-Mean Melt (AMmelt) and Area-Mean Retreat (AMretreat), respectively (Table 1). Thermal forcing profiles are calculated as done for ISMIP6melt and ISMIP6retreat, but are extrapolated across the glacier terminus and an area-mean thermal forcing is calculated instead of pulling from a set depth (Figure C1). A third parameterization (AMberg) accounts for bathymetric obstructions in the same way as ISMIP6melt, but the surface 175 m follows the Gade slope submarine melt mixing line (Gade, 1979; Straneo and Cenedese, 2015) to approximate the influence of iceberg submarine melting on the upper water column (Figure 2). As surface salinity in the fjord remains relatively similar to external water (Figure 2), we leave salinity unchanged from ISMIP6melt, and adjust temperatures accordingly to fit the Gade slope (Table 1). A final modification is made in which waters below freezing temperature are set to the *in-situ* freezing point (Figure C1). In practice, AMberg approximates a lower limit of cooling that occurs through submarine melting, while AMmelt sets the upper bound on surface temperatures attainable for non-iceberg



runs (Figure 2). Therefore, we test a fourth, middle-ground parameterization (AMconst) in which the ISMIP6melt temperature and salinity at 175 m is extrapolated to the surface before calculating the thermal forcing profile (Figure 2; Table 1). For both AMconst and AMberg, the final thermal forcing value used in Equations 1 and 2 is again an area-mean across the glacier face.

The profiles used to calculate AMmelt and AMretreat each had a skill score of $\sim 0.55$ and predicted $\Theta_{\bar{A}}$ within an RMSE of 0.30°C and 1.24°C, respectively; however, the RMSE of AMretreat remained heavily dependent on sill depth (Table 2). As expected, AMmelt was a strong predictor of near-glacier thermal forcing in non-iceberg runs (RMSE = 0.13°C; SS = 0.73), while AMberg performed well in iceberg runs (RMSE = 0.19°C; SS = 0.91). AMconst was an adequate compromise between AMberg and AMmelt, with an RMSE of 0.31°C and a skill score of 0.69 across the entire model ensemble (Table 2). Skill
scores for all parameterizations decreased considerably with successively shallower sills (Table 2).

Encouraged by our model results, we test the efficacy of AMberg with observations using conductivity, temperature, and depth profiles collected by ship (CTDs) and helicopter (XCTDs) within Ilulissat Icefjord and adjacent Disko Bay in August 2014 (Beaird et al., 2017) and August 2019 (Figure 5; Appendix B). Temperature and salinity data from Disko Bay were used to calculate an average external thermal forcing profile analogous to ISMIP6retreat. Following the steps outlined above, a
profile of AMberg was then constructed based on average Disko Bay temperature and salinity profiles. In both years, AMberg reproduced XCTD casts within Ilulissat Icefjord with a mean skill score of 0.95 (Eq. 12), compared to mean ISMIP6retreat and ISMIP6melt skills scores of 0.43 – 0.62 and 0.19 – 0.27, respectively. Agreement between AMberg and observations was particularly pronounced in the upper water column where vast volumes of meltwater are released from the fjord's dense and extensive mélange (Enderlin et al., 2014). However, observed thermal forcing below sill depth was higher than predicted by
AMberg and ISMIP6melt (Figure 5), which is in contrast to our Ilulissat Icefjord-style simulations (Figure 2e). We interpret this discrepancy to be the result of temporally varying conditions in Ilulissat Icefjord (which we neglect in our steady-state simulations) that reflect previously warmer conditions in Disko Bay.

While AMberg demonstrates a marked improvement from other parameterizations in capturing the cooling effects of icebergs, it does so by sacrificing accuracy in non-iceberg runs. Therefore, the true merit of AMberg would be its tandem use with
AMmelt and some *a priori* knowledge or likelihood estimate of iceberg presence, whereby AMmelt could be used in fjords with few icebergs and AMberg used where icebergs are prevalent (here called AMfit). Such a tactic could predict near-glacier thermal forcing within an error of 0.15°C and a skill score of 0.82 across all model runs (Table 2), but is particularly effective in S400 (RSME = 0.10°C; SS = 0.91) and S250 runs (RSME = 0.09°C; SS = 0.83).

### 4.3   Impact of Thermal Forcing Parameterizations on ISMIP6 Submarine Melt Rates

Our modeling indicates that the magnitude of thermal forcing used in ice sheet models, and thus parameterized submarine melt rates, is highly sensitive to the depth range where thermal forcing is defined. In a given simulation, $\Theta_{\bar{z}}$, $\Theta_{\bar{A}}$, and $\Theta_{\overline{gl}}$ differed by 0.38 – 1.12°C, which can translate to a difference of $> 200$ m yr$^{-1}$ in submarine melt rates parameterized by Eq. 2 (Figure 6). (Note our use of Eq. 2 here is slightly outside its intended use, which original relied on thermal forcing defined between 200 m depth and the grounding line.) An even greater thermal forcing range existed between the seven parameterizations
tested in this paper (Table 1). Within a given run, thermal forcing parameterizations differed by 0.90 – 2.26°C (Figure 6).



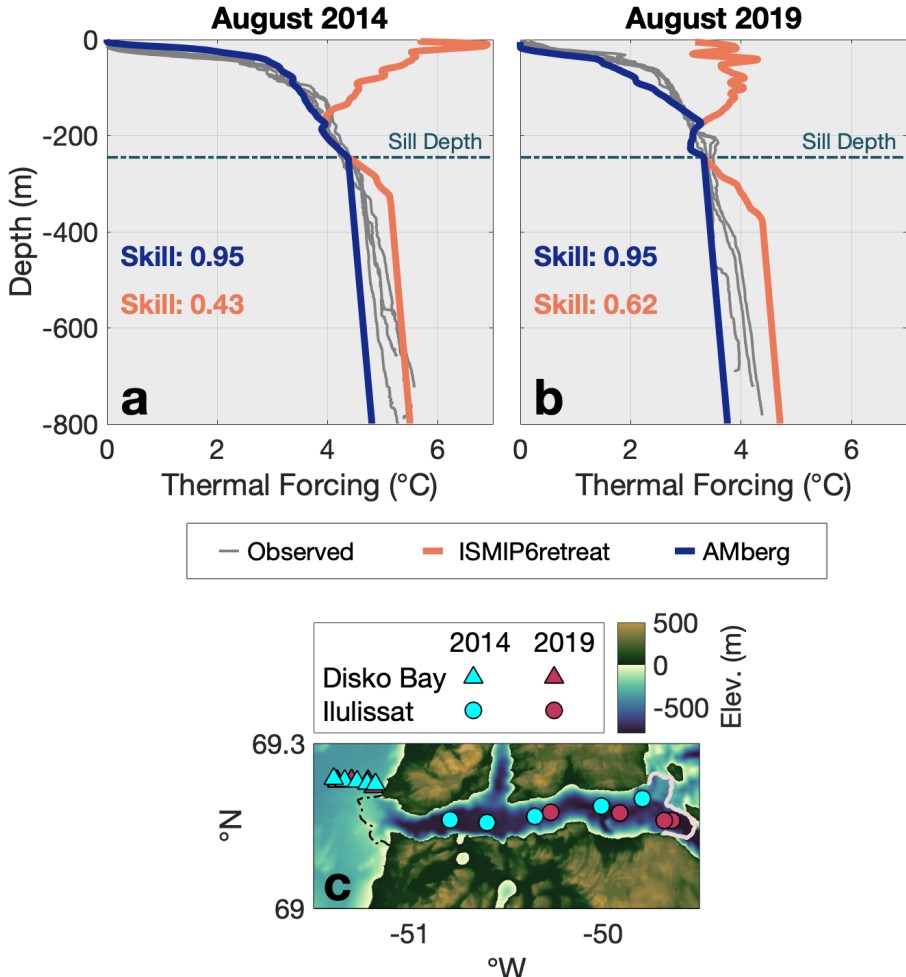

**Figure 5.** Thermal forcing profiles (gray) observed within Ilulissat Icefjord via XCTD casts in **(a)** August 2014 and **(b)** August 2019, plotted with the profiles used to calculate ISMIP6retreat (orange) and AMberg (dark blue). The ISMIP6retreat profile is equivalent to external forcing conditions in Disko Bay. Skill score for each parameterization profile is also provided. Horizontal dashed line signifies the maximum depth (245 m) of the Ilulissat Icefjord entrance sill. **(c)** Bathymetric map rendering the locations of XCTD casts within Ilulissat Icefjord (circles) used in **a-b**, as well as XCTD and CTD casts in Disko Bay (triangles) used to provide external forcing conditions. Solid rose line depicts the Sermeq Kujalleq terminus position in July 2014 (Goliber et al., 2022), and the dashed black line outlines the location of the entrance sill.

Depending on the thermal forcing method used, submarine melt rates parameterized by Eq. 2 could therefore differ by 91 – 617 m yr$^{-1}$ for a single fjord (Figure 6). The difference between parameterizations was greatest in S100 summer runs, largely due to the pronounced bathymetric effects that ISMIP6retreat and AMretreat do not account for. Also notable is the difference between thermal forcing parameterizations in our Ilulissat Icefjord-style low resolution runs (Figure 6), which due to the high

subglacial discharge of Sermeq Kujalleq would contribute an uncertainty in parameterized submarine melt rates of > 300 m



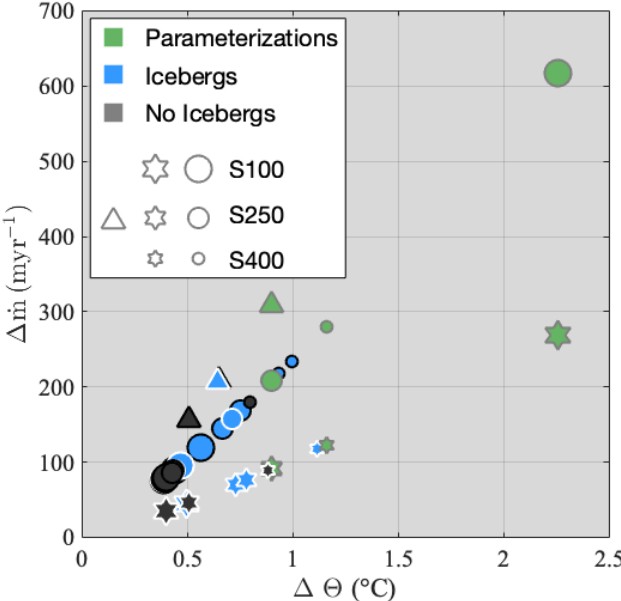

**Figure 6.** The difference between $\Theta_{\bar{z}}$, $\Theta_{\bar{A}}$, and $\Theta_{\overline{gl}}$ in each simulation ($\Delta\Theta$), plotted with the resulting range of parameterized submarine melt rates ($\Delta\dot{m}$), as defined by Eq. 2. Iceberg runs are light blue and non-iceberg runs are gray. White-edged markers depict runs with sheet plumes, stars represent winter ($Q_{sg} = 10 \text{ m}^3 \text{ s}^{-1}$) simulations, and triangles indicate low resolution ($Q_{sg} = 1000 \text{ m}^3 \text{ s}^{-1}$) runs. Marker size varies by sill depth. Green markers depict the difference between all seven thermal forcing parameterizations explored in this paper for each sill depth and subglacial discharge group, again plotted against the resultant range of parameterized submarine melt rates.

yr$^{-1}$. Sermeq Kujalleq single-handedly accounts for $10\%$ of GrIS mass loss (Smith et al., 2020) and dominates uncertainty in mass balance projections of central-West Greenland (Goelzer et al., 2020). Therefore, this wide range between thermal forcing parameterizations could signify substantial uncertainty in the projected terminus position of Sermeq Kujalleq, and thus mass loss projections for the GrIS as a whole.

**4.4 Remaining Uncertainty Associated with Sill-Driven Mixing**

While AMberg effectively parameterizes the two largest sources of uncertainty in predicting near-glacier thermal forcing – bathymetric obstruction and iceberg meltwater – it does not account for the modification of deep water though sill-driven reflux, which was found to be significant in this study and in previous work (Hager et al., 2022; Davison et al., 2022; Kajanto et al., 2022; Bao and Moffat, 2023). The influence of reflux is difficult to discern through EOF analysis, although multiple 435 lines of evidence highlight its importance in shallow silled fjords. First, heat reflux has the potential to exceed even the greatest iceberg heat fluxes, and is responsible for the cooling of deep water ($88\%$ of the water column in S100 runs) by up to $\sim 0.6°$C (Figure 4). Second, the RMSE of each parameterization is greatest in S100 runs, in which more than $37\%$ of outflow water is refluxed. This is true despite correcting for the bathymetric obstruction of external water in most parameterizations. Thus,




there must be an additional source of error related to sill depth, which can only be explained through the cooling of deep water
through the reflux of iceberg meltwater and subglacial discharge.

Deep water cooling from sill-driven mixing is not expected to be important in fjords with sills deeper than iceberg keels
or the summer terminal plume depth; however, reflux is likely influential in a number of critical glacial fjord systems. Only
10 individual glaciers are responsible for the majority of GrIS mass loss (Fahrner et al., 2021), three of which, Kakiffaat
Sermiat, Sverdrup Gletsjer, and Sermeq Kujalleq, terminate in fjords with sills $\lesssim 250$ m deep (as compared to grounding line
depths of $400 - 800$ m) (Morlighem et al., 2017). It is in these fjords that we expect heat reflux to significantly influence near-
glacier temperatures. Indeed, recent modeling of Ilulissat Icefjord indicates the sill-driven reflux of iceberg meltwater cools
near-glacier water by $0.2°C$ (Kajanto et al., 2022), a result shared by our Ilulissat Icefjord-style low resolution runs.

Although the updated parameterizations presented in this paper greatly decrease error compared to existing ISMIP6 meth-
ods, incorporation of sill-driven mixing could further reduce error in shallow silled fjords, such as the Ilulissat Icefjord-Sermeq
Kujalleq system. We anticipate such improvements would require the use of box-models that contain representations of ice-
berg melting, subglacial discharge, and reflux. The strong linear dependence of $\alpha^r_{out}$ on $Q'_{fw}/Q_{fw}$ (Figure 4b) indicates reflux
fractions can be accurately estimated from the vertical distribution of freshwater fluxes in the water column, without any knowl-
edge of tidal forcing. Thus, Eq. 11 could be used within a box-model to predict reflux, assuming the model can approximate
freshwater fluxes throughout the water column.

## 5   Conclusions

In summary, we have tested the accuracy of common ocean thermal forcing parameterizations across a wide range of local
forcing scenarios and fjord geometries, and identified fjord bathymetry and iceberg melt-driven cooling as the two greatest
sources of error when translating regional water properties to tidewater glacier termini. Even a simple adjustment for fjord
bathymetry, as done in the ISMIP6 submarine melt implementation, can predict grounding line thermal forcing within an
average error of $0.2°C$; however, without accounting for bathymetry parameterizations may overpredict thermal forcing by at
least $2°C$ in shallow silled fjords.

Neither ISMIP6 parameterization could reliably reproduce an entire profile of near-glacier thermal forcing in our simulations
due to iceberg-driven cooling of the upper water column. To address this concern, we made a simple correction to the ISMIP6
submarine melt implementation that can predict the near-glacier thermal forcing profiles of our iceberg-laden runs, as well as
observed thermal forcing within Ilulissat Icefjord, to a skill score of $> 0.90$. While promising, this approach sacrifices accuracy
in fjords with few icebergs, and thus would be best utilized in conjunction with an iceberg prevalence prediction method, which
could apply the iceberg parameterization only to fjords where iceberg melt is significant.

Our modeling also highlights the sensitivity of thermal forcing – and therefore the parameterized submarine melt rate – to
the depth range that is chosen. If a depth range that incorporates surface layers is chosen, for example, a calving front area
mean thermal forcing, then iceberg-driven cooling becomes important and can reduce the prescribed thermal forcing by $1°C$ in
many Greenland fjords. Regardless, the AMfit parameterization presented here can predict entire near-glacier thermal forcing



profiles with an average skill score of 0.82 across all bathymetries and iceberg concentrations tested in our simulations. As such, AMfit could be used to accurately parameterize $\Theta_{\bar{z}}$, $\Theta_{\bar{A}}$, $\Theta_{\overline{gl}}$, or an entire $\Theta(z)$ profile, depending on the requirements and capabilities of the ice sheet model. This result highlights the need for an iceberg prevalence prediction method to be developed

and implemented in the next generation of ice sheet models.

Additional improvements to the parameterizations presented here could take the form of a box-model that can effectively represent sill-driven reflux. While such a tactic is outside the scope of this paper, we have identified a strong linear relationship between reflux and the ratio of freshwater released below sill depth (Eq. 11), which we suggest could be used to efficiently estimate reflux in box-models of Greenland fjords. We emphasize that such models also need to accurately parameterize

iceberg melting and subglacial discharge plumes, as well as incorporate local fjord bathymetry. Still, the revised thermal forcing parameterizations presented in this paper are an improvement to existing methods, while their simplicity makes them relatively straightforward to implement.

*Data availability.* Abbreviated model output and statistics, as well as input and restart files for each simulation, are available at https://doi.org/10.5281/zenodo.7826386 (Hager et al., 2023). Full model output and processing code are available upon request. CTD data

from Ilulissat Icefjord are also available at https://doi.org/10.5281/zenodo.7826386 (Hager et al., 2023).

## Appendix A: TEF and Efflux/Reflux Theory

Efflux/reflux theory quantifies the net effect of mixing without the need to resolve individual mixing processes (Cokelet and Stewart, 1985). Effectively, efflux/reflux transports can be thought of as the vertical equivalent of the horizontal TEF budget (MacCready et al., 2021). In TEF terms, mass and volume conservation require:

$$\alpha_{in}^r Q_{in}^o + \alpha_{out}^e Q_{out}^g = Q_{out}^o$$
$$\alpha_{in}^r S_{in}^o Q_{in}^o + \alpha_{out}^e S_{out}^g Q_{out}^g = S_{out}^o Q_{out}^o$$


$$\alpha_{in}^e Q_{in}^o + \alpha_{out}^r Q_{out}^g = Q_{in}^g$$
$$\alpha_{in}^e S_{in}^o Q_{in}^o + \alpha_{out}^r S_{out}^g Q_{out}^g = S_{in}^g Q_{in}^g$$

(A1)

where superscripts $o$ and $g$ denote the TEF transports on the oceanward and glacierward sides of the mixing zone, respectively, and subscripts indicate whether the transport is inflowing (glacierward) or outflowing (oceanward). Superscripts on $\alpha$ signify the percent of the inflowing or outflowing layer that is refluxed ($\alpha_{in,out}^r$) or effluxed ($\alpha_{in,out}^e$) at the sill (Figure A1). The



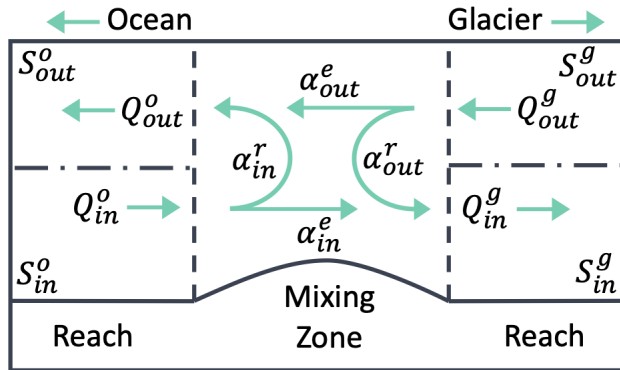

**Figure A1.** Schematic illustrating each variable in efflux/reflux theory (Eq. A1).

solutions to Eq. A1 are:

$$\alpha_{in}^r = \frac{Q_{out}^o}{Q_{in}^o} \frac{S_{out}^o - S_{out}^g}{S_{in}^o - S_{out}^g} \qquad \alpha_{out}^e = \frac{Q_{out}^o}{Q_{out}^g} \frac{S_{in}^o - S_{out}^o}{S_{in}^o - S_{out}^g}$$

(A2)

$$\alpha_{in}^e = \frac{Q_{out}^g}{Q_{out}^o} \frac{S_{in}^g - S_{out}^g}{S_{in}^o - S_{out}^g} \qquad \alpha_{out}^r = \frac{Q_{in}^g}{Q_{out}^g} \frac{S_{in}^o - S_{in}^g}{S_{in}^o - S_{out}^g}.$$

Mass and volume conservation also require:

$$\alpha_{in}^r + \alpha_{in}^e = 1$$

(A3)

$$\alpha_{out}^r + \alpha_{out}^e = 1$$

and

$$S_{out}^g \leq S_{out}^o < S_{in}^o$$

(A4)

$$S_{out}^g < S_{in}^g \leq S_{in}^o.$$

TEF budgets are not exact and even at steady-state some drift still occurs within the mixing zone; therefore, we make minor adjustments to the TEF transports ensuring Eqs. A1, A3, and A4 are satisfied before solving Eq. A2 (e.g., MacCready et al., 2021; Hager et al., 2022), but the resultant error on $\alpha_{out}^r$ was $\leq 0.04\%$ of the reported value.

## Appendix B: Collection and Processing of Hydrographic Data

Hydrographic properties were observed via helicopter-based XCTD casts within Ilulissat Icefjord (four profiles in August
2014 and five in August 2019) and by shipboard CTD casts in Disko Bay (four in August 2014; five in August 2019). In both



years, an additional XCTD profile was obtained in Disko Bay for comparison with CTD casts. Shipboard CTD casts were conducted using an RBR XR-620 in 2014 and an RBRconcerto in 2019. All casts were averaged into 1 m bins and smoothed using a running mean filter with a 10 m window (comparable to our model vertical resolution) before undertaking the analysis described in Section 4.2. Further details about the 2014 field campaign are described in Beaird et al. (2017).



**Appendix C: Additional Figures and Tables**

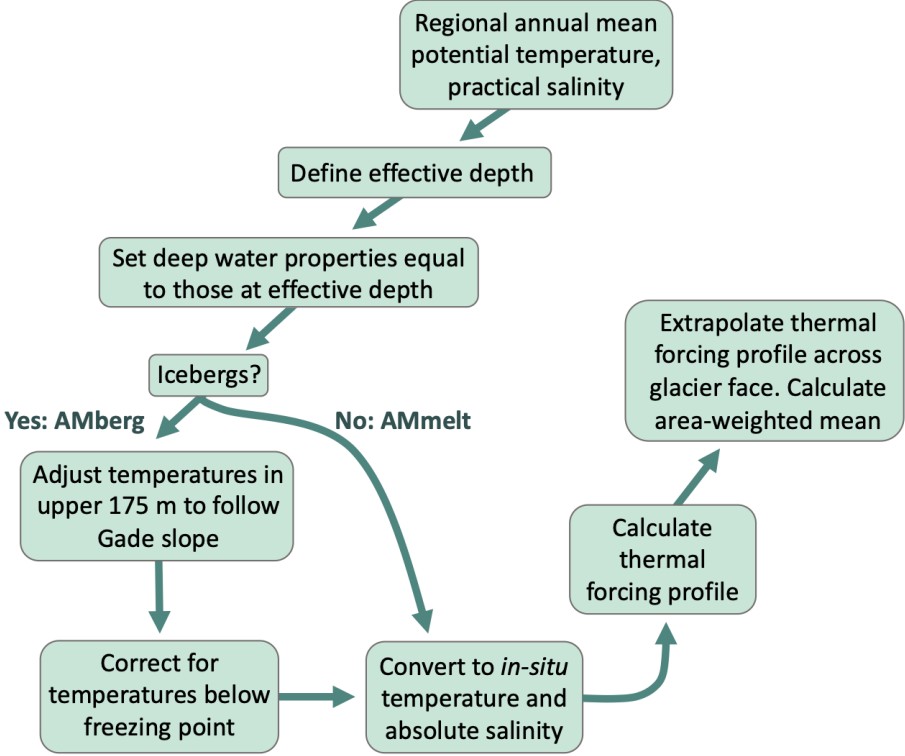

**Figure C1.** Flow chart illustrating the step-by-step process for computing AMberg and AMmelt.





**Table C1.** Fjord horizontal resolution ($Res_H$), maximum tidal velocity at the western open boundary ($U_T$), subglacial discharge ($Q_{sg}$) and plume type (L is line-plume; HC is half-conical plume), sill depth ($Z_{sill}$), maximum iceberg keel depth ($Z_{berg}^{max}$), maximum iceberg concentration ($SA_{berg}^{max}$), minimum iceberg concentration ($SA_{berg}^{max}$), area-weighted mean near-glacier thermal forcing ($\Theta_{\bar{A}}$), and reflux percent ($\alpha_{out}^r$) for all runs.

| $Res_H$ (m) | $U_T$ cm s$^{-1}$ | $Q_{sg}$ (m$^3$ s$^{-1}$) | $Z_{sill}$ (m) | $Z_{berg}^{max}$ (m) | $SA_{berg}^{max}$ (%) | $SA_{berg}^{min}$ (%) | $\Theta_{\bar{A}}$ (°C) | $\alpha_{out}^r$ (%) |
|---|---|---|---|---|---|---|---|---|
| **Summer** | **Icebergs** | | | | | | | |
| 200 | 0.5 | 300 (HC) | 400 | 300 | 25 | 5 | 4.4 | 0.045 |
| 200 | 0.5 | 300 (HC) | 400 | 300 | 75 | 5 | 4.4 | 0.12 |
| 200 | 0.5 | 300 (HC) | 250 | 300 | 25 | 5 | 3.9 | 0.039 |
| 200 | 0.5 | 300 (HC) | 250 | 300 | 75 | 5 | 3.8 | 0.024 |
| 200 | 0.5 | 300 (L) | 250 | 300 | 75 | 5 | 3.8 | 24 |
| 200 | 0.5 | 300 (L) | 100 | 300 | 75 | 5 | 2.1 | 70 |
| 200 | 0.5 | 300 (HC) | 100 | 300 | 25 | 5 | 2.4 | 48 |
| 200 | 0.5 | 300 (HC) | 100 | 300 | 75 | 5 | 2.2 | 53 |
| 500 | 0.5 | 1000 (HC) | 250 | 400 | 90 | 90 | 3.7 | 17 |
| 500 | 0.5 | 1000 (L) | 250 | 400 | 90 | 90 | 3.7 | 24 |
| **Summer** | **No Icebergs** | | | | | | | |
| 200 | 0.5 | 300 (HC) | 400 | | | | 4.7 | 0.12 |
| 200 | 0.5 | 300 (HC) | 250 | | | | 4.1 | 16 |
| 200 | 0.5 | 300 (L) | 250 | | | | 4.1 | 41 |
| 200 | 0.7 | 300 (HC) | 250 | | | | 4.1 | 0.22 |
| 200 | 0.3 | 300 (HC) | 250 | | | | 4.1 | 8.8 |
| 200 | 0.5 | 300 (HC) | 100 | | | | 2.9 | 37 |
| 200 | 0.5 | 300 (L) | 100 | | | | 2.9 | 73 |
| 200 | 0.7 | 300 (HC) | 100 | | | | 2.9 | 37 |
| 200 | 0.3 | 300 (HC) | 100 | | | | 2.9 | 38 |
| 500 | 0.5 | 1000 (HC) | 250 | | | | 4 | 0.019 |
| **Winter** | **Icebergs** | | | | | | | |
| 200 | 0.5 | 10 (L) | 400 | 300 | 75 | 5 | 4.2 | 26 |
| 200 | 0.5 | 10 (L) | 250 | 300 | 75 | 5 | 3.8 | 0.024 |
| 200 | 0.5 | 10 (L) | 100 | 300 | 75 | 5 | 2.0 | 68 |
| 500 | 0.5 | 10 (L) | 250 | 400 | 90 | 90 | 3.6 | 16 |
| **Winter** | **No Icebergs** | | | | | | | |
| 200 | 0.5 | 10 (L) | 400 | | | | 4.4 | 46 |
| 200 | 0.5 | 10 (L) | 250 | | | | 3.9 | 66 |
| 200 | 0.5 | 10 (L) | 100 | | | | 2.9 | 75 |



*Author contributions.*  Alexander Hager designed and conducted the model experiments, performed the analysis, and wrote the paper. David Sutherland secured funding and provided guidance throughout the research process. Donald Slater contributed his expertise in ISMIP6 ocean thermal forcing parameterizations that advanced the direction of research. All authors contributed to editing the manuscript.

*Competing interests.*  The contact author has declared that none of the authors has any competing interests.

*Acknowledgements.*  This project was funded by NSF grant 2020447 from the Office of International Science and Engineering, as well as NSF grant 2023269 from the Arctic Natural Sciences program within the Office of Polar Programs. Donald Slater acknowledges support from NERC Independent Research Fellowship NE/T011920/1.



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
