# Peer review of "Local forcing mechanisms challenge parameterizations of ocean thermal forcing for Greenland tidewater glaciers"

_EGUsphere, 2023_

## Referee Comment (RC1)

**Review Hager et al. 2023**

Michele Petrini

**General comment:**

In this paper, Hager et al. address a topic which is extremely relevant for the future sea-level contribution of the Greenland ice sheet (and as such, well within the scope of this journal): evaluating the ability of ISMIP6 ocean thermal forcing parameterizations to predict thermal forcing at tidewater glacier termini. This is accomplished through experiments with the MITgcm, using a set of idealized Greenland fjords and ocean boundary conditions, and parametrised subglacial discharge, glacier submaring melting (IcePlume package) and icebergs (IceBerg package). Sensitivity tests are designed by varying tidal amplitudes, subglacial discharge, iceberg coverage, and bathymetry. Incorporating and assessing the impact of iceberg melting in fjord simulations represents an important innovation, and the approach and methodology used by the authors is sound, although I think some additional clarifications and reorganization are needed in the Methodology section (see minor specific comments below). The authors indicate that the bathymetric control on the intrusion of Atlantic water into the fjords is the primary control on near-glacier thermal forcing, followed by iceberg submarine melting. It is found that grounding line thermal forcing varied by 2.9 $^{\circ}$C across all simulations and is heavily dependent on the depth of bathymetric sills in relation to the Polar-Atlantic Water thermocline. The authors highlight that using a simple adjustment for fjord bathymetry, the ISMIP6 submarine melt implementation is able to predict grounding line thermal forcing within 0.2 $^{\circ}$C. Finally, Hager et al. introduce new parameterizations accounting for iceberg-driven cooling, which accurately predicted interior fjord thermal forcing profiles in both iceberg-laden simulations and observations from Ilulissat Icefjord. The results are presented in a very clear and structured way, and fully support the authors' conclusions, which are extremely relevant for the ice-sheet modelling community.

In view of this, I recommend this work for publication, and I only have some minor comments which are listed below.

**Specific comments:**

1) It would be good to have some additional text (either in the main text or in the supplementary) explaining the choice on the simulation length and output averaging choice (L96-99). From what I read in the text, I am left with two main questions: (1) why water properties stop evolving after different amount of time in different simulations (2) as simulations are meant to represent a seasonal evolution, it is somehow strange to see they are extended up to 2.5 years. I don't expect this to be a major issue, but it would be nice to see an explanation.

2) It is a bit confusing to find the new parametrizations in Table 1 well before they are defined in the text. One simple solution could be to refer to the section where they are introduced in Table 1 (for instance: New Parametrizations (see section xxx);

3) I think table C1 should belong to the main text, as it is extremely informative and widely referenced to. Moreover, in Subsection '2.1 Model setup', I found it not immediately easy to have a broad overview of the differences in each simulation. Including Table C1 in the main text would likely be enough, but also some simple text reorganization could be useful (for instance: the total number of simulations is provided only at the end in L134-135);

**Technical comments/suggestions:**

L29: it could be good to specify/expand to what extent these processes are small scale (spatial and temporal) compared to those in global climate models (and ice-sheet models).

L30: Suggest splitting sentences, e.g., "To date, sea level projections have instead …".

L31: Maybe 'simplified'?

L32: Suggest 'that are large sources of uncertainty'. Also, 'future mean sea levels'.

L87 and elsewhere throughout the text: Suggest either adding South/North/West/East arrays in Fig. 1, or use different naming (e.g., along fjord, across fjord?) as it is not immediately clear where S/N/W/E are.

L105: Maybe explain why significant tidal mixing was expecting, or add a citation?

L243-245: missing reference to Fig./table? Don't know where percentages come from

L280 and formula 12: not sure if this explanation should be moved to the methods section, similarly as subsections 2.3, 2.4, 2.5.

L319: perhaps something like 'its contribution to the variability of near-glacier…'?

L372: Maybe better use 'Such an approach'? Same for later occurences.

L462: 'ISMIP6 parametrizations'.

Figure 2: I am confused by the presence of Qberg and Hberg shadings: what are they (Hberg is introduced only later in Fig. 4.), and are they cited in the text? It is ok to keep them, but at least an explanation in the legend is needed.  Also, there is a typo in the inbox legend, purple line should read ISMIP6melt & AMmelt.

---

## Author Comment (AC1)

**Review Hager et al. 2023**

Michele Petrini

**General comment:**

In this paper, Hager et al. address a topic which is extremely relevant for the future sea-level contribution of the Greenland ice sheet (and as such, well within the scope of this journal): evaluating the ability of ISMIP6 ocean thermal forcing parameterizations to predict thermal forcing at tidewater glacier termini. This is accomplished through experiments with the MITgcm, using a set of idealized Greenland fjords and ocean boundary conditions, and parametrised subglacial discharge, glacier submaring melting (IcePlume package) and icebergs (IceBerg package). Sensitivity tests are designed by varying tidal amplitudes, subglacial discharge, iceberg coverage, and bathymetry. Incorporating and assessing the impact of iceberg melting in fjord simulations represents an important innovation, and the approach and methodology used by the authors is sound, although I think some additional clarifications and reorganization are needed in the Methodology section (see minor specific comments below). The authors indicate that the bathymetric control on the intrusion of Atlantic water into the fjords is the primary control on near-glacier thermal forcing, followed by iceberg submarine melting. It is found that grounding line thermal forcing varied by 2.9 °C across all simulations and is heavily dependent on the depth of bathymetric sills in relation to the Polar-Atlantic Water thermocline. The authors highlight that using a simple adjustment for fjord bathymetry, the ISMIP6 submarine melt implementation is able to predict grounding line thermal forcing within 0.2 °C. Finally, Hager et al. introduce new parameterizations accounting for icebergdriven cooling, which accurately predicted interior fjord thermal forcing profiles in both iceberg-laden simulations and observations from Ilulissat Icefjord. The results are presented in a very clear and structured way, and fully support the authors' conclusions, which are extremely relevant for the ice-sheet modelling community.

In view of this, I recommend this work for publication, and I only have some minor comments which are listed below.

We thank Dr. Petrini for reading our manuscript and providing a thorough and positive review, which we believe will be beneficial to the paper. We have provided responses to Dr. Petrini's concerns and have outlined how we will edit the manuscript accordingly.

**Specific comments:**

1) It would be good to have some additional text (either in the main text or in the supplementary) explaining the choice on the simulation length and output averaging choice (L96-99). From what I read in the text, I am left with two main questions: (1) why

water properties stop evolving after different amount of time in different simulations (2) as simulations are meant to represent a seasonal evolution, it is somehow strange to see they are extended up to 2.5 years. I don't expect this to be a major issue, but it would be nice to see an explanation.

Water properties stopped evolving after all fjord water had been flushed, which occurred from the surface down the water column. Sill depth was the primary control on flushing time, because shallower silled simulations had a larger volume of water below sill depth, which additionally renewed at a slower rate than water above the sill. Subglacial discharge and plume type (which controls entrainment and displacement of deep water in the plume), as well as tidal forcing, also influenced flushing time. As simulations evolved at different rates, it was important to run simulations to steady-state to ensure each simulation had fully responded to its unique forcing conditions and we were thus comparing apples to apples. Although in reality it is unlikely Greenland glacial fjords are ever at steady-state, this is a tacit assumption of the ISMIP6 parameterizations that we want to remain consistent to. We will add 1-2 sentences at Line 98 explaining 1) why steady-state is important to our simulations, and 2) why simulations reached steady-state at different rates.

The averaging of output over the last 10 days of our simulation was done to remove any influence of tides, internal waves, etc. from our results. As the runs are at steady-state, averaging over this time period will not impact our results other than to remove noise generated by tides. Again, we can elaborate on this point at Line 99.

It is a bit confusing to find the new parametrizations in Table 1 well before they are defined in the text. One simple solution could be to refer to the section where they are introduced in Table 1 (for instance: New Parametrizations (see section xxx);

Thank you for this suggestion – we will add "(see Section 4.2)" to the Table 1 caption

2)  I think table C1 should belong to the main text, as it is extremely informative and widely referenced to. Moreover, in Subsection '2.1 Model setup', I found it not immediately easy to have a broad overview of the differences in each simulation. Including Table C1 in the main text would likely be enough, but also some simple text reorganization could be useful (for instance: the total number of simulations is provided only at the end in L134-135);

Thank you for making this point. Table C1 will be added to the Methods section of the main text. We will also add a new short paragraph at Line 99 that will provide an overview of the forcings used and the total number of runs before diving into the rest of the model setup.

**Technical comments/suggestions:**

L29: it could be good to specify/expand to what extent these processes are small scale (spatial and temporal) compared to those in global climate models (and ice-sheet models).

Thanks for the suggestion. This sentence will be changed to:

"However, such processes are too small scale (~1 m to ~1 km length scales at hourly to seasonal timescales) to be resolved in global climate models (grid resolution of ~30-60 km at annual timescales; e.g., Watanabe et al., 2010; Golaz et al., 2019)."

L30: Suggest splitting sentences, e.g., "To date, sea level projections have instead …".

This change will be made.

L31: Maybe 'simplified'?

This change will be made.

L32: Suggest 'that are large sources of uncertainty'. Also, 'future mean sea levels'.

These changes will be made in the next version of the manuscript.

L87 and elsewhere throughout the text: Suggest either adding South/North/West/East arrays in Fig. 1, or use different naming (e.g., along fjord, across fjord?) as it is not immediately clear where S/N/W/E are.

Thank you for the suggestion – a compass rose will be added to Figure 1a

L105: Maybe explain why significant tidal mixing was expecting, or add a citation?

The expectation of significant tidal mixing in shallow-silled glacial fjords is based on previous work by Hager et al. (2022) and Bao and Moffat (2023). We will add additional background with citations to this sentence, as well as add a sentence at line 169 describing why we are interested in sill-driven mixing.

L243-245: missing reference to Fig./table? Don't know where percentages come from

Thank you for pointing this out. Iceberg and submarine meltwater fluxes will be added to Table C1, which will now be Table 1 in the main text (see above).

L280 and formula 12: not sure if this explanation should be moved to the methods section, similarly as subsections 2.3, 2.4, 2.5.

The formula and surrounding text will be moved to the end of Section 2.2. For consistency, we will also introduce the use of root mean square errors in Section 2.2.

L319: perhaps something like 'its contribution to the variability of near-glacier...'?

Thanks for the suggestion - this change will be made in the next version of the text.

L372: Maybe better use 'Such an approach'? Same for later occurences.

These changes will be made in the next version of the text.

L462: 'ISMIP6 parametrizations'.

We think this should be kept as is because it is preceded by "neither".

Figure 2: I am confused by the presence of Qberg and Hberg shadings: what are they (Hberg is introduced only later in Fig. 4.), and are they cited in the text? It is ok to keep them, but at least an explanation in the legend is needed. Also, there is a typo in the inbox legend, purple line should read ISMIP6melt & AMmelt.

Qberg and Hberg are included to illustrate the extent of iceberg melting with depth, particularly in relation to sill depth and variability in upper water column. Qberg and Hberg are first introduced in Section 2.4 (Equation 6). Additional references to Figure 2 will be made at lines 224 and 236, and Lines 215-216 will be changed to "…with only minor variability occurring when iceberg keels extended below sill depth (see $Q_{berg}$ and $H_{berg}$ profiles in Figure 2) …" to draw specific attention to the Qberg and Hberg profiles.

We will also change the last sentence of the caption to "The vertical distribution of iceberg freshwater fluxes (Qberg ) and heat fluxes (Hberg ) are provided in a-c and d-f, respectively, to depict the depth of iceberg melt relative to sill depth and profile variability."

Thank you also for spotting the typo in the legend – this will be fixed in the revision.

---

## Author Comment (AC2)

**Review of "Local forcing mechanisms challenge parameterization of ocean thermal forcing for Greenland tidewater glaciers" by Hager et al.**

Hager et al. present ocean model simulations with MITgcm for an idealized domain and use those to test the accuracy of melt parameterisations for Greenland fjords as they are used in large-scale projections. This is a relevant work as ocean-driven retreat of glaciers is one of the important processes driving Greenland mass loss and of interest for publication in TC. I suggest some modifications to the analysis and presentation as detailed below to improve the accuracy and understanding of the work.

We thank the reviewer for their reading of the manuscript and providing suggestions that will improve its quality. We have incorporated as many suggestions as possible into the new version of the manuscript. However, the authors respectfully disagree with some of the reviewer's comments; namely, there seems to be confusion on the difference between submarine melt parameterizations and thermal forcing parameterizations, as well as our statistical approach for comparing thermal forcing parameterizations. In cases of disagreement, the authors do their best to find a compromising solution when possible, or provide reasoning for the maintenance of the original text.

**General comments:**

- **Structure:** the structure of the manuscript could be improved as at the moment it is not clearly going towards one aim, which makes it hard to read. Information is spread into several places, e.g., the ISMIP melt parameterisations are introduced in the introduction, the new ones in parts in the Methods 2.2, in the discussion in Section 4.2. You could make the thermal forcing parameterizations your central point and move it earlier. In addition, you should introduce all thermal forcing parameterizations explicitly, i.e., giving their equations, in the methods in Section 2.2. Then you can validate them against the model simulations in the results and discuss their caveats and benefits in the Discussion. Ideally, you can end with a recommendation.

   The central point of this manuscript is the testing of ISMIP6 thermal forcing parameterizations. This is explicitly stated in Lines 8-9, 69-71, 451-454 and implicitly throughout. The purpose of the ISMIP6 melt/retreat parameterizations provided in the Introduction (Eqs. 1 and 2) are to provide context for both ISMIP6 thermal forcing parameterizations and to set the stage for the rest of the paper. The thermal forcing parameterizations introduced in Methods 2.2 do not originate from this study, but are two separate methods previously used by ISMIP6 to calculate the thermal forcing terms in Eqs. 1 and 2. We include the ISMIP6 thermal forcing parameterizations in the Methods because we directly use them in our study, whereas the ISMIP6 submarine melt/retreat paramterizations are only used for context. After extensive testing of the ISMIP6 thermal forcing parameterizations in the Methods and Results sections, we found they were inadequate to accurately extrapolate far-field ocean thermal forcing to the near-glacier region, so we thus introduce possible alternatives in the Discussion section. This step can only be accomplished following the results from the ISMIP6 thermal forcing parameterizations, as alternatives would not be needed if ISMIP6 parameterizations had performed well. Based on our additional testing, we encourage the use of AMfit in lines 471-474, as it is the most accurate thermal forcing parameterization tested; however, ice sheet models do not yet have the capability to predict iceberg prevalence, so we refrain from making a hard recommendation of this method until other capabilities of ice sheet models improve. Additionally, in lines 476-482 we recommend possible avenues for the development of a fjord-scale box model that could further improve coupling between global climate and ice sheet models.

Equations for the ISMIP6 thermal forcing parameterizations are provided in Lines 139 and 151. However, most of the differences between parameterizations are accomplished through step-by-step data manipulation, and do not cleanly lend themselves to written equations. We therefore find the combination of equations and written descriptions of our parameterizations, as done in Sections 2.2, 4.2, and Figure C1, to be the most effective way to communicate this information. This is the same strategy as done in other papers that use thermal forcing parameterizations (e.g., Slater et al., 2019; Slater et al., 2020; Morlighem et al., 2019; and Cowton et al. 2018). We acknowledge that the Gade Slope (Line 381) is not well defined in our manuscript, and will include this equation in the next version of the manuscript.

**Experimental design / results:** At the moment you are comparing apples and oranges for the different parameterizations: the AMmelt/ISMIP6melt and AMretreat/ISMIP6retreat parameterizations are evaluated by comparing theta_gl, while the AMberg, AMconst and AMfit parameterization are evaluated with the profile. This makes it hard to actually see how much AMberg improves over AMmelt (there is a lot about the importance of the iceberg melt in the document, but the actual effect on melt rates remains unclear, as it influences mainly the upper layers). I suggest that you compare all parameterizations with respect to all three quantities theta_gl, theta_z, theta_A as well as the corresponding melt rates through equation (2), and also compare all to the measurements (Fig 5). Best would be to summarize results for all parameterizations in one table / figure. Otherwise, it is not clear how you rank the importance of processes (section 4.1).

The use of different thermal forcing metrics comes from ISMIP6 experiments, because each ISMIP6 thermal forcing parameterization is designed to predict either theta_z (as is the case ISMIP6retreat) or theta_gl (as is the case for ISMIP6melt). By comparing the parameterizations only to their intended metric, we believe we are in fact avoiding an apples to oranges comparison. For example, drawing a comparison between theta_z and ISMIP6melt would be asking the parameterisation to do something it was never intended to do.

The two separate methods used in ISMIP6 experiments to parameterize submarine melting (Eqs. 1 and 2) rely on two different definitions of near-glacier thermal forcing. Equation 1 relies on ISMIP6retreat, which was originally developed by Slater et al. (2019) to parameterize $\theta_{\bar{z}}$ (a depth average thermal forcing between 200-500m depth). Conversely, Equation 2 relies on ISMIP6melt, which was originally developed by Morlighem et al. (2019) to parameterize $\theta_{gl}$ (grounding line thermal forcing). When testing the accuracy of these thermal forcing parameterizations, we compare each only to its intended thermal forcing metric. In this way, we ensure that we are testing its accuracy in a manner consistent with the original intent of the parameterization.

As discussed in Lines 350-375, using a depth-dependent scalar to define near-glacier thermal forcing creates uncertainty in thermal forcing parameterizations. We therefore developed new area-mean parameterizations in Section 4.3, which are an attempt to minimize this uncertainty. The root mean square error of each of these new parameterizations (AMberg, AMmelt, AMconst, AMretreat, AMfit) is a comparison to the area-mean near-glacier thermal forcing ($\theta_{\bar{A}}$) in our simulations, as that is the metric these parameterizations were intended to represent. AMretreat, AMmelt, and ISMIP6retreat are never compared to $\theta_{gl}$, as this would be inconsistent with their intended purpose.

All profiles used to create each parameterization were assessed by their ability to parameterize a full near-glacier thermal forcing profile. As discussed in Lines 276-280, this is done because there is still an ongoing glaciological debate over which thermal forcing definition is most influential on ice dynamics. As discussed by the reviewer, the results from all parameterizations are already summarized in Table 3. The metric that each parameterization is compared to when calculating the root mean square error ($\theta_{gl}$, $\theta_{\bar{z}}$, or $\theta_{\bar{A}}$) is detailed in Table 3 and the Table 3 caption. The authors appreciate the reviewer's suggestion to compare all parameterizations to the observations of Ilulissat Icefjord, and will include this information in Table 3.

The submarine melt parameterizations used in ISMIP6 and within the MITgcm are known to be inaccurate (e.g., Jackson et al., 2020), and we therefore avoid reporting absolute melt rates from Eq. 2 or the MITgcm. As the purpose of this paper is just to test the accuracy of the thermal forcing parameterizations, we limit this discussion to the range of melt rates provided by Eq. 2 when using the various thermal forcing parameterizations (Section 4.3). The authors will add a sentence clarifying this point in Section 4.3.

- **Generalization of results**: in your ocean model runs you use one background forcing and one idealized geometry – how much do your results depend on this? You should at least discuss this caveat.

We agree that this is an important point – thank you for bringing it up. We designed our experiments with one constant background forcing to be compatible with the methodology of ISMIP6 experiments. In ISMIP6, the Greenland coast is divided into seven regions in which temperatures and salinities are annually averaged, so that all modeled glaciers within a given region experience the same offshore ocean conditions. Our experiments thus emulate this approach by imposing the same "regional" background forcing in all of our simulations. In effect, we created an arbitrary ISMIP6 "region", then tested how much fjord conditions may vary within that region based on local forcing mechanisms (Lines 73-35).

Previous studies point to sill-driving mixing (and thus sill depth) as a primary mechanism for local water transformation and control on fjord water properties (e.g., Ebbesmeyer and Barnes, 1980; Cokelet and Stewart, 1985; Hager et al., 2022, Bao and Moffat, 2023), while fjord width, length, and depth are not expected to greatly influence water properties, just circulation (e.g., Carroll et al., 2018). Thus, we chose to focus on sill depth as the primary geometric constraint by using three different idealized geometries: S100, S250, and S400. These geometries were chosen to span the depth range of the Atlantic-Polar Water thermocline, which is a ubiquitous feature around Greenland. As we draw our conclusions from the relative depths of the sill and thermocline (not absolute depths), we do not anticipate this choice will impact our results. We acknowledge this wasn't fully clear in the manuscript, so we will add a sentence to Section 2.1 explaining our choice to focus on sill-driven mixing, instead of other geometric constraints.

We will also add a sentence at Line 461 akin to: "While we have made attempts to compare our idealized results against observations of multiple Greenland fjords, we anticipate some variability when applied to realistic fjord geometries and forcing" to highlight the comparison of our results to observations in multiple locations in Greenland (Lines 340-349, lines 396-407, and Figure 5), but acknowledge some uncertainty exists when applying these parameterizations to realistic fjords.

**Specific comments:**

Abstract:

- Line 13: What the 2.9°C refer to is unclear, maybe rather give the maximum modification that the TD experiences.

  We believe it is already clear what the 2.9C refers to, so would like to leave this sentence as is, but perhaps the reviewer could expand on what is unclear?

- Line 15-17: It's unclear if your parameterisation includes bathymetry?

  Thank you making this point – we will add the word "additionally" to this sentence to make clear that the iceberg parameterization also includes the adjustment for bathymetry discussed in the previous sentence.

Introduction:

- line 31: Morlighem et al., 2019 is no projection, Jourdain et al., 2020, introduces parameterisations for Antarctica, so neither citation really fits to your sentence

  Both citations here are in reference to "simplifying parameterizations of oceanic boundary conditions" and not "sea level rise projections." To the authors' knowledge, Morlighem et al. (2019) is the original study that uses a thermal forcing parameterization that has a bathymetric adjustment, similar to ISMIP6melt. Jourdain et al. (2020) introduces ocean thermal forcing parameterizations used in the Antarctic ISMIP6 experiments. As this sentence is a general statement about parameterizations used in sea level rise projections, and is not specific to Greenland, we feel this citation is justified here.

  We acknowledge the original reading of this sentence was confusing in regards to the citations, so we will change the sentence to read:

  "… and instead, sea level rise projections have relied on poorly-validated simplifying parameterizations of ocean boundary conditions in ice sheet models, such as those developed in Morlighem et al. (2019), Jourdain et al. (2020), and Slater et al. (2019). These parameterizations create large sources of uncertainty when predicting future mean sea levels …"

- line 31: Seroussi et al. is for Antarctica, the citation does not fit here.
  This sentence is a general statement about the impact of thermal forcing parameterizations on the uncertainty of sea level rise projections and is not specific to Greenland. We therefore feel this citation is justified here.

- line 40: Smith et al., 2020 presents satellite observations of thickness changes, it does not link them to the ocean forcing, the citation does not seem to fit here.

  Smith et al. (2020) presents satellite observations of thickness changes, but importantly, also pins changes in ice thickness to specific atmospheric and/or ocean forcing. A primary result from this paper is that ocean warming is responsible for modern day ice loss from Antarctica, while

ice loss from Greenland is the combined result of heightened atmospheric and ocean forcing. Specifically, our purpose for citing this paper in Line 40 is to highlight their conclusion that: "…the combination of increased surface melt and warmer ocean temperatures has led to the enhanced submarine melting of submerged glacier termini and has allowed more rapid calving by reducing the presence of rigid mélange in the fjords, each of which have increased glacier velocities and ice discharge into the ocean."

- Equation (1) here glacier front changes are directly linked to frontal melt changes, however, this misses out changes in ice dynamics: a glacier terminus could stay in the same position for higher melting when the ice discharge increases at the same time (at least for a while). This seems to be missing some physics?

  Yes, this is a crude parameterization of ocean-driven glacier retreat and is undoubtedly missing important physics, as is acknowledged in the paper describing the parameterization – Slater et al. (2019). Nonetheless, this is one of the two parameterizations for Greenland frontal ablation used by ISMIP6 experiments, and we only include it here only to provide context for thermal forcing parameterization it uses. It is outside the scope of this paper to evaluate the legitimacy of this equation, as we are solely concerned with thermal forcing parameterizations.

- Explicitly state somewhere that you do not evaluate melt parameterizations, just the thermal forcing aspect. And state clearly, that the ISMIP parameterisations underlies a thermal forcing parameterisation, that the resulting melt is relevant, however, this is still open and here always done using the equation (2, except for the retreat parameterisation in ISMIP6).

  The purpose of this paper – to test the accuracy of the thermal forcing parameterizations – is stated in Lines 69-71 and elaborated on in Lines 75–78. We include the ISMIP6 submarine melt parameterization here only for context and it is outside the scope of this paper to assess the validity of this equation. To avoid confusion, we will add a sentence at Line 778 stating: "This paper focuses solely on thermal forcing parameterizations and makes no attempt to test the validity of Eqs. 1 and 2, which are provided here only for context."

- My understanding is that equation (2) is mainly used to put the importance of thermal driving differences in the context to melt rates and not suggested as a valid melt parameterisation? If this is correct, state it.

  Please see response to above comment.

Methods:

- line 121: Where was the background velocity implemented?

  This is a parameter within IcePlume and is needed to drive melt across the glacier face. We will restructure the wording of this sentence to make it more evident where this is implemented.

- What about sea ice in MITgcm?

  Thank you for making this point, as this is one local forcing mechanism we ignore. To limit unconstrained parameters, we do not explicitly include sea ice in our experiments. However, the

influence of sea ice melt is likely captured to some degree by the parameterization of surface iceberg melting through the IceBerg Package. As iceberg depths are prescribed using an inverse power law size frequency distribution, a vast majority of icebergs are very shallow and could be thought of as representing sea ice interspersed among larger icebergs. One caveat to this reasoning is that IceBerg does not account for brine rejection caused from sea ice formation. While we do not expect this process influence near-glacier thermal forcing in deep fjords that flush frequently, it is possible sea ice could be an important factor in some shallow fjords. We will add a sentence to this effect at Line 133.

- line 155: do you want a new paragraph for the sentence "We compare.."

  Thanks for the suggestion - we will make this sentence its own paragraph in the next version of the manuscript.

- line 157: Does "modeled area-mean" mean that it is averaged over the entire depth? And above, is the TD at the grounding line the one from the lowest cell?

  Yes, the area-mean is an average across the entire area of the glacier face (both vertically and horizontally). The grounding line is located at the lowest cell of the glacier face. We can insert clarification to this sentence to make these definitions clearer.

- Table 1: Define better exactly how the thermal forcing is calculated (e.g., which grid cells are used, just the closest to the calving front or are they averaged? How is this handled with different resolutions?).

  As with other "near-glacier" metrics, near-glacier thermal forcing is a 10-day average of the two rows of cells closest to the glacier face (as described in Lines 97-98). This is the same for both resolutions.

- in general, I miss more motivation for your methods, e.g., why do you want to quantify sill-driven mixing? Why do you use three thermal forcing metrics (and not just one)?

  The focus on sill-driven mixing is motivated by previous studies (e.g., Ebbesmeyer and Barnes, 1980; Cokelet and Stewart, 1985; Hager et al., 2022; Bao and Moffat, 2023) that show fjord mixing is primarily restricted to sill regions. A clarifying sentence will be added to the beginning of Section 2.3.

  The use of three thermal forcing metrics was done to be consistent with the original goals of ISMIP6retreat and ISMIP6melt. ISMIP6retreat was originally designed by Slater et al. (2020) to parameterize theta_z, while ISMIP6melt was designed by Morlighem et al. (2019) to parameterize theta_gl. We therefore test ISMIP6retreat and ISMIP6melt by comparing to theta_z and theta_gl in our model, respectively, to ensure we are comparing equivalent quantities. As described in Lines 358–375, theta_A was then developed in this paper as an alternative to theta_z and theta_gl that is sensitive to other processes not captured by the original ISMIP6 parameterizations. The authors will reword Lines 155 – 157 in the original text to make this clearer.

- furthermore, I miss a motivation and explanation for the newly introduced melt parameterisations in the method.

  The parameterizations introduced in the methods are the thermal forcing parameterizations used by Eqs. 1 and 2 to drive frontal ablation, and are not new melt parameterizations. We feel this is appropriately described in Lines 136 – 138, Line 153, and throughout Section 2.2, but are open to feedback on how to clarify this distinction.

Results:

- Line 210: Not sure where exactly you find the grounding line average salinity in the Figure? Is is simply the deepest value (at -800m)?

  Grounding line water properties are taken from near-glacier cells at the base of the glacier, here at -800m. This clarification can be added to Line 98 in the original text.

- line 215: "..when iceberg keels extend… or subglacial discharge … below sill depth" – from the figure 2, this seems to be true for sill depth of -250 and -100m. How can you draw the logical conclusion that this is linked to the keel depth and vertical extend of the plume from this figure?

  In S400 runs, no icebergs extend below sill depth and water properties are entirely homogenous below sill depth. In S250 runs, only two runs have significant variability below sill depth; these are the two low resolution runs with iceberg keel depths extending to 400 m. Variability in these profiles only occurs in the upper 400 m and coincides with the input of iceberg melt water (and associated heat sinks) shown with $Q_{berg}$ and $H_{berg}$ in panels b and e. In all iceberg S100 runs, keels extend below sill depth and thus contribute to cooling of the entire water column (in combination with sill-driven reflux). Additional variability seems to coincide with the terminal plume depths shown in black and white triangles.

  This sentence states that variability below sill depth only occurs when iceberg keels extend below sill depth or when subglacial plumes reach neutral buoyancy below sill depth. This holds true for all runs, and water below sill depth remains homogenous in all runs where this is not the case.

- line 219-221: this is hard to see from Figure 2. At least in panel (e) it looks like there might be blue triangles left and right of black triangles (and the lines intersect above of -200m).

  Thanks for the suggestion - Figure 2 will be rearranged to make these symbols more visible. A reference to Table C1 will also be added, which also contains the same information.

- line 224: again, this refers to the middle and right columns, or how can this be seen more precisely in the figure?

  Figure 2 will be rearranged so that each panel is enlarged and this pattern more visible.

- line 237: the third EOF "depicts temperature variability coincident with the terminal depth of subglacial plumes" – I am not sure this is is very clear, e.g., the lower terminal plume depth

around -400 m does not coincide with a change in the temperature profile? Why does this EOF not represent the reflux?

The bottom cluster of terminal plume depths does coincide with a modulation in the shape of the third EOF mode, as do the approximate depths of the upper two clusters of terminal plume depths. In both this study and in Davison et al. 2022, reflux uniformly cools/warms the water column below sill depth (and should not alter water properties above sill depth), and thus we do not believe the third EOF mode could represent this process. While the authors acknowledge that the physical interpretation of EOFs modes can be ambiguous, we feel subglacial discharge is the most plausible explanation for third EOF mode. However, as we are least confident with this physical interpretation compared to the other EOF modes, we will add language at line 237 to reflect this uncertainty.

- line 243-247: where are the absolute numbers? Can you add a table containing them?

  Freshwater fluxes for each run will be added to Figure C1 and moved into the main paper.

- line 248 – 250: this is simply because of the latent heat required to melt the icebergs, or?

  Yes, as is explained in Lines 310 – 312. We feel this is better served as a discussion point, because it is an interpretation of the data.

- Figure 2: Are the profiles from the center of the calving front or are they averaged over the calving face? I would mention earlier on that the columns are for the different sill depth, e.g., add this as titles to the columns. The figure is quite dense, you could help the reader by indicating what features they should look at in the figure. E.g. for the sentence in lines 212-214 "However, water properties…" you could add in the end ".. for S100 runs (compare the blue and black triangles indicating the depth-averaged thermal driving in the ocean simulations across the three lower panels). Same for lines 216, explain how the reader can see that "iceberg keels extend beyond sill depth" and "subglacial plumes reach neutral buoyancy". Same for the next sentence. It looks like some triangles are missing, e.g., there are no black triangles in panel (f)?

  As described in Lines 92-32, all "near-glacier" output is an average of all cells within two rows of the glacier face. We can change the wording of this sentence to make this clearer. Sill depth column titles were intentionally left out because the figures are already dense and it was the intent that the horizontal dashed line depicting sill depth could make this distinction. We will move up the explanation of these lines and the difference between the columns higher up in the caption so this is immediately evident to the reader.

  We will change "(Figure 2)" in Line 214 to "(black/blue triangles in Figure 2d-f)".

  All black triangles are accounted for, but some overlap others. As discussed above, we will rearrange and enlarge Figure 2 to make these more visable.

  Lines 215-217 will be changed to "…with only minor variability occurring when iceberg keels extended below sill depth (see $Q_{berg}$ and $H_{berg}$ profiles in Figure 2) subglacial discharge plumes reached neutral buoyancy below sill depth (this most often occurs with line-plumes; see black/white triangles in Figure 2a-c)." Additionally, the last line of the caption will be changed to

"The vertical distribution of iceberg freshwater fluxes (Qberg ) and heat fluxes (Hberg ) are provided in a-c and d-f, respectively, to depict the depth of iceberg melt relative to sill depth and profile variability."

- Figure 4b: What does this mean that there is higher reflux with higher freshwater input at depth?

  The greater the portion of freshwater input that enters the system below sill depth, the greater the portion of water that is refluxed at the entrance sill.

- equation 12: what is the motivation for this "skill score" definition, is this something commonly used?

  This is a commonly used metric to compare modeled profiles of ocean/atmospheric properties to observations and was originally developed in Willmott (1982). We will include a citation to this paper in Line 280.

Discussion:

- I would move part of the discussion to the results, e.g., the definition of the new parameterisations for thermal driving, how this is translated into melt.

  The main purpose of this paper is to evaluate the current ISMIP6 methods for parameterizing thermal forcing and identify the primary sources of error. The additional parameterizations introduced in Section 4.2 and the discussion on melt rate uncertainty (Section 4.3) should therefore be treated as an exploration of possible improvements to the current ISMIP6 methods and what impact this could have on ISMIP6 melt rates. Thus, we feel these sections are better suited as discussion points.

- Section 4.1: you are comparing unlike things here as you are using for 1. the average thermal driving as the relevant quantity, while in 2.-4. your relevant quantity is the variability in the thermal driving profile. If you want to list the processes "in order of importance", I suggest that you think about what defines their importance (relevant quantity is resulting basal melt rate, temperature profile or the average temperature) and then compare them with respect to this quantity.

  This is a good point and alludes to one of the main difficulties in establishing a thermal forcing parameterization that is useful for ice sheet models. As discussed in Lines 358-375, the best method for defining an ocean thermal forcing metric that is relevant to glacier frontal ablation processes is still a topic of ongoing debate. It is unclear if frontal ablation can be accurately determined solely from grounding line thermal forcing, mean thermal forcing, or an entire profile. At the same time, current submarine melt parameterizations cannot be corroborated by direct observations of melt at glacier termini. Therefore, there is no straightforward relevant quantity of interest for thermal forcing parameterizations. Instead, Section 4.1 determines the level of importance of each mechanism primarily based on its influence on full profile variability (including the translation of profiles that occurs between sill depth groups, which is equivalent to Theta_A), thus capturing any potentially relevant quantity of interest. When possible, we

then use additional lines of evidence, such as heat fluxes, to further corroborate our ranking of important processes.

Accordingly, the authors will make the following changes to the manuscript:

1. Add a sentence at the beginning of Section 4.1 explaining why direct comparison is challenging and explaining our reasoning for comparison metrics
2. As discussed previously, we will reword Lines 155-157 to explain why multiple thermal forcing metrics exist and the difficulty of establishing a relevant quantity of interest
3. Add a sentence to the beginning of Section 4.3 explaining why we don't compare our results to absolute submarine melt rates.

- lines 350 and following: is this caveat ("the dependence on specific depth when calculating thermal forcing") not the same caveat as discussed in the paragraph above, i.e., that sills are highly relevant for thermal forcing in the fjords?

  This sentence is referring to the ISMIP6 practice of defining Theta_z and Theta_gl at specific depths (Theta_z is only defined between 200-500 m and Theta_z is only defined at the grounding line). We can reword this sentence to clarify this point.

- give the equations for the parameterisations, e.g., how exactly follows the AMberg the Gade line (what ambient water masses do you assume to mix with, how much mixing occurs, see line 380)?

  An equation for the Gade slope will be added to Line 380.

- line 386: "*iceberg* melting" instead of "submarine melting"?

  This will be changed to "iceberg melting"

- Figure 5: ISMIP6retreat label should be AMretreat, or (this is also mixed up in the text)? Please add the other thermal driving parameterisations as well, i.e.,. AMmelt, AMconst as well as dots for the ISMIP6 ones. How well do they perform?

  This should be ISMIP6retreat, although the profiles used for ISMIP6retreat and AMretreat are the same. Other parameterizations can be added to this plot.

- line 405: ISMIP6melt is not on the figure 5, AMretreat shows higher temperatures. The difference could also stem from other reasons than "temporally varying conditions", i.e., horizontal variability in the sill and ice conditions...

  ISMIP6melt will be added to the figure. All observed profiles (gray) have warmer temperatures below sill depth than are predicted by the profile for AMberg (blue).

  We see no evidence of meaningful horizontal temperature gradients between the sill-driven mixing zone and glacier face in our modeling, regardless of iceberg distribution, subglacial discharge, etc. Observations of glacial fjords generally support this claim (e.g., Straneo et al.,

2012; Mortensen et al., 2014, Moffat et al., 2018, Gladish et al., 2015). Therefore, the most likely explanation is that the warm water observed at depth is a remnant of warmer water that had previously entered the fjord earlier in the summer. We will add a sentence in Section 3.1 describing the lack of strong horizontal gradients in our simulations, and will refer to that sentence in Line 405 when providing our interpretation.

- Figure 6: Difference between each theta and what? The far-field theta/boundary conditions? How are sub-shelf melt rates calculated with (2) when using a thermal driving profile? Non-iceberg runs are black? I think also the green markers show the thermal forcing parameterisations differences in thermal driving relative to the boundary conditions / ISMIP6retreat case? Don't you model melt rates with MITgcm and IcePlume, why don't you compare to those as well?

  Figure 6 depicts the difference between the various thermal forcing definitions (Theta_z, Theta_A, and Theta_gl) within each of our simulations, as is also described in Lines 415-418. The word "difference" in the caption may thus be better described as a "range", and we will change the wording accordingly.

  Thermal forcing parameterizations were computed as described in the text and the resultant value is used as Theta in Eq. 2. The scalar values provided by each parameterization, not the profiles used to calculate the scalars, are used in Eq. 2.

  Non-iceberg runs are dark gray. The gray box in the legend was intended to be the same shade of gray as the markers, but there was a bug in the code. This will be fixed.

  The green markers depict the range of all seven thermal parameterizations explored in the paper, grouped by sill depth and subglacial discharge. Again, we will change the word "difference" to "range" to make this more evident.

  As discussed previously, we avoid comparing our results to absolute submarine melt rates, because melt parameterizations are not yet reliable. We will include this point in Section 4.3.

- line 417: how much is the 200m/yr in relative terms (i.e., how large are the melt rates overall)?

  This is a difficult comparison to make, as direct observations of tidewater glacier submarine melt rates do not exist in Greenland to our knowledge, and model submarine melt parameterizations are known to be inaccurate. Nonetheless, an uncertainty of 200 m/yr stemming just from ocean thermal forcing parameterizations (in addition to the added uncertainty within the formulation of the melt parameterization) is likely to be significant for some glaciers, particularly when coupled to other feedback mechanisms (e.g., calving processes, etc.).

- line 420: within a given run, theta_z, theta_A and theta_gl calculated by the forcing parameterisations differed…(or how did you estimate the difference)?

  This sentence refers to the seven thermal forcing parameterizations explored in this paper and referred to in the previous sentence (ISMIP6melt, ISMIP6retreat, AMberg, etc.), and not theta_z,

theta_A, or theta_gl, which are metrics calculated from the model to test the parameterizations. The difference is the range of thermal forcing values given by the parameterizations.

- line 434: Could it be that the reflux is hard to get from the EOF because it is linked to the bathymetry and you removed that in your EOF analysis?

  That is likely the case here, as was discussed in Lines 238-241.

- line 484: "reduces error in *thermal driving profiles* compared to ismip6 estimates"..

  Thank you for this suggestion – Line 448 will be changed to "Although the updated parameterizations present in this paper greatly reduce thermal forcing error compared to …"

- line 461: add "in shallow silled fjords *in our idealised simuations*".

  Thank you for this suggestion. This sentence will be changed to "… however, without accounting for bathymetry, parameterizations overpredict thermal forcing by at least 2◦C in our idealized shallow-silled simulations."

- Add that it remains an open question which theta to use, or how to translate the profile into melt rates.

  This is done in detail in Lines 358-375 and touched on again in Lines 472-474. A new sentence will be added at Line 475 to emphasize this as an importance direction of future research.

- I miss a discussion of the next steps (develop and evaluate melt parameterizations) and caveats (idealized model domain, only one background forcing, comparing to one fjord,...).

  Multiple next steps are discussed in the conclusion. These include (1) development of an iceberg prevalence prediction method so that AMfit can be used accurately (Lines 474 – 475), and (2) the development of a box-model parameterization that includes reflux (Lines 476 – 480). As noted above, we will also add a sentence highlighting the importance of determining which thermal forcing metric to use in melt parameterizations. The development and evaluation of melt/retreat parameterizations is a related but separate area of research.

  Lines 340-349, lines 396-407, and Figure 5 are aimed at broadening the results of our idealized simulations to observations of multiple Greenland fjords. Nonetheless, we will add a sentence akin to "While we have made attempts to compare our idealized results against observations of multiple Greenland fjords, we anticipate some variability when applied to realistic fjord geometries and forcing" at Line 461.

  The choice to use one background temperature/salinity forcing was made to be consistent with the design of ISMIP6 experiments, and temperature/salinity profiles at the open boundaries were chosen to represent conditions surrounding Greenland. The feature of importance in boundary condition is the depth of the Atlantic-Polar water thermocline in relation to the sill depth. While the depth of this thermocline may change throughout Greenland, we base our findings only on its depth relative to sill depth, and thus, we do not anticipate this choice impacting our results.

Appendix:

- State what the abbreviation TEF stands for.

  Please see Line 160 for a definition of TEF.

- line 489: what do you mean with volume conservation? In general, mass, energy and momentum are conserved, but volume might change with density (temperature, salinity, pressure) changes. Please explain.

  The TEF framework is based on the assumption that the mass and volume entering the mixing zone must equal the mass and volume exiting the mixing zone.

- Equation A4: How do these follow from "mass and volume conservation"? Is it rather that you assume salinity on the glacier side is lower because of mixing with melt water?

  Yes, TEF is specifically designed for estuaries where there is a freshwater source at the head of the estuary. Therefore, the freshest layer must be the outward flowing (upper) layer on the glacierward side of the mixing zone, and the densest layer must be the inward flowing (lower) layer on the ocean side of the mixing zone. Then due to mixing across the sill, the remaining layers must have a density somewhere between these two. This is described in detail within the sources cited in this section.

- Figure C1: This is the algorithm for AMfit, right? The definition of effective depth could be repeated here, or you could point to the relevant location in the methods.

  This is the step-by-step process for computing AMberg and AMmelt, which when implemented in iceberg and non-iceberg laden fjords, respectively, make AMfit. We can point to Section 2.2 where effective depth is defined.

- Table C1: Please add theta_gl and theta_z here as well.

  Thanks for the suggestion - Theta_gl and Theta_z will be added to the table

---

## Author Response (AR1)

**Review Hager et al. 2023**

Michele Petrini
**General comment:**

In this paper, Hager et al. address a topic which is extremely relevant for the future sea-level contribution of the Greenland ice sheet (and as such, well within the scope of this journal): evaluating the ability of ISMIP6 ocean thermal forcing parameterizations to predict thermal forcing at tidewater glacier termini. This is accomplished through experiments with the MITgcm, using a set of idealized Greenland fjords and ocean boundary conditions, and parametrised subglacial discharge, glacier submaring melting (IcePlume package) and icebergs (IceBerg package). Sensitivity tests are designed by varying tidal amplitudes, subglacial discharge, iceberg coverage, and bathymetry. Incorporating and assessing the impact of iceberg melting in fjord simulations represents an important innovation, and the approach and methodology used by the authors is sound, although I think some additional clarifications and reorganization are needed in the Methodology section (see minor specific comments below). The authors indicate that the bathymetric control on the intrusion of Atlantic water into the fjords is the primary control on near-glacier thermal forcing, followed by iceberg submarine melting. It is found that grounding line thermal forcing varied by 2.9 °C across all simulations and is heavily dependent on the depth of bathymetric sills in relation to the Polar-Atlantic Water thermocline. The authors highlight that using a simple adjustment for fjord bathymetry, the ISMIP6 submarine melt implementation is able to predict grounding line thermal forcing within 0.2 °C. Finally, Hager et al. introduce new parameterizations accounting for icebergdriven cooling, which accurately predicted interior fjord thermal forcing profiles in both iceberg-laden simulations and observations from Ilulissat Icefjord. The results are presented in a very clear and structured way, and fully support the authors' conclusions, which are extremely relevant for the ice-sheet modelling community.
In view of this, I recommend this work for publication, and I only have some minor comments which are listed below.

We thank Dr. Petrini for reading our manuscript and providing a thorough and positive review, which we believe will be beneficial to the paper. We have provided responses to Dr. Petrini's concerns and have outlined how we will edit the manuscript accordingly. Line numbers of the revisions refer to the revised manuscript, except where otherwise specified.

**Specific comments:**

1) It would be good to have some additional text (either in the main text or in the supplementary) explaining the choice on the simulation length and output averaging choice (L96-99). From what I read in the text, I am left with two main questions: (1) why water properties stop evolving after different amount of time in different simulations (2) as simulations are meant to represent a seasonal evolution, it is somehow strange to see they are extended up to 2.5 years. I don't expect this to be a major issue, but it would be nice to see an explanation.

Water properties stopped evolving after all fjord water had been flushed, which occurred from the surface down the water column. Sill depth was the primary control on flushing time, because shallower silled simulations had a larger volume of water below sill depth, which additionally renewed at a slower rate than water above the sill. Subglacial discharge and plume type (which controls entrainment and displacement of deep water in the plume), as well as tidal forcing, also

influenced flushing time. As simulations evolved at different rates, it was important to run simulations to steady-state to ensure each simulation had fully responded to its unique forcing conditions and we were thus comparing apples to apples. Although in reality it is unlikely Greenland glacial fjords are ever at steady-state, this is a tacit assumption of the ISMIP6 parameterizations that we want to remain consistent to. In the revised manuscript, we added an additional sentence at Lines 103 – 104 explaining 1) why steady-state is important to our simulations, and 2) why simulations reached steady-state at different rates. We also added "depending on fjord flushing time" to the parentheses in the preceding sentence (Line 102).

The averaging of output over the last 10 days of our simulation was done to remove any influence of tides, internal waves, etc. from our results. As the runs are at steady-state, averaging over this time period will not impact our results other than to remove noise generated by tides. In the revision, we inserted the phrase "to remove the influence of tides or internal waves" to this sentence (Line 105).

It is a bit confusing to find the new parametrizations in Table 1 well before they are defined in the text. One simple solution could be to refer to the section where they are introduced in Table 1 (for instance: New Parametrizations (see section xxx);

Thank you for this suggestion – we added "(see Section 4.2)" to the table caption (now Table 2).

2) I think table C1 should belong to the main text, as it is extremely informative and widely referenced to. Moreover, in Subsection '2.1 Model setup', I found it not immediately easy to have a broad overview of the differences in each simulation. Including Table C1 in the main text would likely be enough, but also some simple text reorganization could be useful (for instance: the total number of simulations is provided only at the end in L134-135);

Thank you for making this point. Table C1 is now Table 1 in main paper. We also moved the total number of simulations from the end of Section 2.1 to Line 107 in the revised text, along with a list of local forcing mechanisms tested.

**Technical comments/suggestions:**

L29: it could be good to specify/expand to what extent these processes are small scale (spatial and temporal) compared to those in global climate models (and ice-sheet models).
Thanks for the suggestion. This sentence has been changed to (Lines 29-31):
"However, such processes are too small scale (~1 m to ~1 km length scales at hourly to seasonal timescales) to be resolved in global climate models (grid resolution of ~30-60 km at annual timescales; e.g., Watanabe et al., 2010; Golaz et al., 2019)."

L30: Suggest splitting sentences, e.g., "To date, sea level projections have instead …".

Original sentence split into two sentences (Line 31)

L31: Maybe 'simplified'?

Changed to "simplified" (Line 32)

L32: Suggest 'that are large sources of uncertainty'. Also, 'future mean sea levels'.

These changes have been made at Lines 33-34 of revised text

L87 and elsewhere throughout the text: Suggest either adding South/North/West/East arrays in Fig. 1, or use different naming (e.g., along fjord, across fjord?) as it is not immediately clear where S/N/W/E are.

Thank you for the suggestion – a compass rose was added to Figure 1a

L105: Maybe explain why significant tidal mixing was expecting, or add a citation?

The expectation of significant tidal mixing in shallow-silled glacial fjords is based on previous work by Hager et al. (2022) and Bao and Moffat (2023). The following changes were made to the manuscript:
1.  Lines 115-116 in the revised manuscript were reworded and citations added to give context for why tidal mixing may be important in shallow-silled fjords.
2.  Additional sentence at beginning of Section 2.3: "Sill-driven mixing is a primary control on circulation and water properties in shallow-silled glacial fjords in Alaska and Patagonia (Hager et al., 2022; Bao and Moffat, 2023) and may additionally influence near-glacier thermal forcing in Greenland." (Lines 186 - 187)

L243-245: missing reference to Fig./table? Don't know where percentages come from

Thank you for pointing this out. Iceberg and glacier submarine meltwater fluxes were added to Table C1, which is now Table 1 in main text. As per RC2's comments, theta_z and theta_gl have also been added to the table. Some rows/columns have subsequently been rearranged. In order to fit expanded table and caption on one page, variable definitions were moved to the text throughout Sections 2.1 and 2.2. Two references to Table 1 were also added to Lines 238 and 242.

L280 and formula 12: not sure if this explanation should be moved to the methods section, similarly as subsections 2.3, 2.4, 2.5.

This paragraph has been added to the end of Section 2.2. The use of root mean square errors is also now introduced in this paragraph (Lines 177 – 184).

L319: perhaps something like 'its contribution to the variability of near-glacier...'?

Thanks for the suggestion - this has been changed to "While subglacial discharge certainly affects near-glacier thermal forcing, its contribution to variability of near-glacier thermal forcing (both inter- and intra-seasonal) is overshadowed by the influence of icebergs." (Lines 341 – 342)

L372: Maybe better use 'Such an approach'? Same for later occurences.

Changed at Lines 396 and 502

L462: 'ISMIP6 parametrizations'.

We think this should be kept as is because it is preceded by "neither".

Figure 2: I am confused by the presence of Qberg and Hberg shadings: what are they (Hberg is introduced only later in Fig. 4.), and are they cited in the text? It is ok to keep them, but at least an explanation in the legend is needed. Also, there is a typo in the inbox legend, purple line should read ISMIP6melt & AMmelt.

Qberg and Hberg are included to illustrate the extent of iceberg melting with depth, particularly in relation to sill depth and variability in upper water column. Qberg and Hberg are first introduced in Section 2.4 (Equation 6).

Additional references to Figure 2 were inserted at lines 253 and 254. Lines 215-216 of the original manuscript were changed to "...with only minor variability occurring when iceberg keels extended below sill depth (see $Q_{berg}$ and $H_{berg}$ profiles in Figure 2) ..." (Lines 243-244 of the revised text) to draw specific attention to the Qberg and Hberg profiles. Qberg and Hberg have also been added to the Figure 2 legend.

We also changed the last sentence of the caption to "The vertical distribution of iceberg freshwater fluxes (Qberg ) and heat fluxes (Hberg ) are provided in a-c and d-f, respectively, to depict the depth of iceberg melt relative to sill depth and profile variability."

Thank you also for spotting the typo in the legend – this has been fixed in the revision.

**Review of "Local forcing mechanisms challenge parameterization of ocean thermal forcing for Greenland tidewater glaciers" by Hager et al.**

Hager et al. present ocean model simulations with MITgcm for an idealized domain and use those to test the accuracy of melt parameterisations for Greenland fjords as they are used in large-scale projections. This is a relevant work as ocean-driven retreat of glaciers is one of the important processes driving Greenland mass loss and of interest for publication in TC. I suggest some modifications to the analysis and presentation as detailed below to improve the accuracy and understanding of the work.

We thank the reviewer for their reading of the manuscript and providing suggestions that will improve its quality. We have incorporated as many suggestions as possible into the new version of the manuscript. However, the authors respectfully disagree with some of the reviewer's comments; namely, there seems to be confusion on the difference between submarine melt parameterizations and thermal forcing parameterizations, as well as our statistical approach for comparing thermal forcing parameterizations. In cases of disagreement, the authors do their best to find a compromising solution when possible, or provide reasoning for the maintenance of the original text. Line numbers refer to the revised manuscript unless otherwise noted.

**General comments:**

- **Structure:** the structure of the manuscript could be improved as at the moment it is not clearly going towards one aim, which makes it hard to read. Information is spread into several places, e.g., the ISMIP melt parameterisations are introduced in the introduction, the new ones in parts in the Methods 2.2, in the discussion in Section 4.2. You could make the thermal forcing parameterizations your central point and move it earlier. In addition, you should introduce all thermal forcing parameterizations explicitly, i.e., giving their equations, in the methods in Section 2.2. Then you can validate them against the model simulations in the results and discuss their caveats and benefits in the Discussion. Ideally, you can end with a recommendation.

  The central point of this manuscript is the testing of ISMIP6 thermal forcing parameterizations. This is explicitly stated in Lines 8-9, 70-72, 489-491 and implicitly throughout. The purpose of the ISMIP6 melt/retreat parameterizations provided in the Introduction (Eqs. 1 and 2) are to provide context for both ISMIP6 thermal forcing parameterizations and to set the stage for the rest of the paper. The thermal forcing parameterizations introduced in Methods 2.2 do not originate from this study, but are two separate methods previously used by ISMIP6 to calculate the thermal forcing terms in Eqs. 1 and 2. We include the ISMIP6 thermal forcing parameterizations in the Methods because we directly use them in our study, whereas the ISMIP6 submarine melt/retreat paramterizations are only used for context. After extensive testing of the ISMIP6 thermal forcing parameterizations in the Methods and Results sections, we found they were inadequate to accurately extrapolate far-field ocean thermal forcing to the near-glacier region, so we thus introduce possible alternatives in the Discussion section. This step can only be accomplished following the results from the ISMIP6 thermal forcing parameterizations, as alternatives would not be needed if ISMIP6 parameterizations had performed well. Based on our additional testing, we encourage the use of AMfit in lines 505-508, as it is the most accurate thermal forcing parameterization tested; however, ice sheet models do not yet have the capability to predict iceberg prevalence, so we refrain from making a hard recommendation of

this method until other capabilities of ice sheet models improve. Additionally, in lines 511-515 we recommend possible avenues for the development of a fjord-scale box model that could further improve coupling between global climate and ice sheet models.

Equations for the ISMIP6 thermal forcing parameterizations are provided in Lines 152 and 164. However, most of the differences between parameterizations are accomplished through step-by-step data manipulation, and do not cleanly lend themselves to written equations. We therefore find the combination of equations and written descriptions of our parameterizations, as done in Sections 2.2, 4.2, and Figure C1, to be the most effective way to communicate this information. This is the same strategy as done in other papers that use thermal forcing parameterizations (e.g., Slater et al., 2019; Slater et al., 2020; Morlighem et al., 2019; and Cowton et al. 2018). We acknowledge that the Gade Slope was not well defined in our manuscript, and it is now included as Eq. 13 (Line 407) in the revised text.

As per the first reviewer's comments, the revised manuscript now also points the reader to Section 4.2 from the Table 2 caption, so that the reader knows where to look for more information about the new parameterizations. We hope this helps improve the flow of the manuscript and addresses some of the above concerns.

**Experimental design / results:** At the moment you are comparing apples and oranges for the different parameterizations: the AMmelt/ISMIP6melt and AMretreat/ISMIP6retreat parameterizations are evaluated by comparing theta_gl, while the AMberg, AMconst and AMfit parameterization are evaluated with the profile. This makes it hard to actually see how much AMberg improves over AMmelt (there is a lot about the importance of the iceberg melt in the document, but the actual effect on melt rates remains unclear, as it influences mainly the upper layers). I suggest that you compare all parameterizations with respect to all three quantities theta_gl, theta_z, theta_A as well as the corresponding melt rates through equation (2), and also compare all to the measurements (Fig 5). Best would be to summarize results for all parameterizations in one table / figure. Otherwise, it is not clear how you rank the importance of processes (section 4.1).

The use of different thermal forcing metrics comes from ISMIP6 experiments, because each ISMIP6 thermal forcing parameterization is designed to predict either theta_z (as is the case ISMIP6retreat) or theta_gl (as is the case for ISMIP6melt). By comparing the parameterizations only to their intended metric, we believe we are in fact avoiding an apples to oranges comparison. For example, drawing a comparison between theta_z and ISMIP6melt would be asking the parameterisation to do something it was never intended to do.

The two separate methods used in ISMIP6 experiments to parameterize submarine melting (Eqs. 1 and 2) rely on two different definitions of near-glacier thermal forcing. Equation 1 relies on ISMIP6retreat, which was originally developed by Slater et al. (2019) to parameterize $\theta_{\bar{z}}$ (a depth average thermal forcing between 200-500m depth). Conversely, Equation 2 relies on ISMIP6melt, which was originally developed by Morlighem et al. (2019) to parameterize $\theta_{gl}$ (grounding line thermal forcing). When testing the accuracy of these thermal forcing parameterizations, we compare each only to its intended thermal forcing metric. In this way, we ensure that we are testing its accuracy in a manner consistent with the original intent of the parameterization.

As discussed in Lines 373-399, using a depth-dependent scalar to define near-glacier thermal forcing creates uncertainty in thermal forcing parameterizations. We therefore developed new area-mean parameterizations in Section 4.3, which are an attempt to minimize this uncertainty. The root mean square error of each of these new parameterizations (AMberg, AMmelt, AMconst, AMretreat, AMfit) is a comparison to the area-mean near-glacier thermal forcing ($\theta_{\bar{A}}$) in our simulations, as that is the metric these parameterizations were intended to represent. AMretreat, AMmelt, and ISMIP6retreat are never compared to $\theta_{gl}$, as this would be inconsistent with their intended purpose.

All profiles used to create each parameterization were assessed by their ability to parameterize a full near-glacier thermal forcing profile. As discussed in Lines 373-398, this is done because there is still an ongoing glaciological debate over which thermal forcing definition is most influential on ice dynamics. As discussed by the reviewer, the results from all parameterizations are already summarized in Table 3. The metric that each parameterization is compared to when calculating the root mean square error ($\theta_{gl}$, $\theta_{\bar{z}}$, or $\theta_{\bar{A}}$) is detailed in Table 3 and the Table 3 caption. The authors appreciate the reviewer's suggestion to compare all parameterizations to the observations of Ilulissat Icefjord, and Figure 5 of the revised manuscript now includes the skill score of each parameterization when used with observations of Ilullisat Icefjord.

The submarine melt parameterizations used in ISMIP6 and within the MITgcm are known to be inaccurate (e.g., Jackson et al., 2020, Sutherland et al., 2019), and we therefore avoid reporting absolute melt rates from Eq. 2 or the MITgcm. As the purpose of this paper is just to test the accuracy of the thermal forcing parameterizations, we limit this discussion to the range of melt rates provided by Eq. 2 when using the various thermal forcing parameterizations (Section 4.3). However, we have added a new short paragraph to the beginning of Section 4.3 that explains why we chose to compare only relative melt rates, instead of absolute melt rates. (Lines 441 - 446).

- **Generalization of results**: in your ocean model runs you use one background forcing and one idealized geometry – how much do your results depend on this? You should at least discuss this caveat.

We agree that this is an important point – thank you for bringing it up. We designed our experiments with one constant background forcing to be compatible with the methodology of ISMIP6 experiments. In ISMIP6, the Greenland coast is divided into seven regions in which temperatures and salinities are annually averaged, so that all modeled glaciers within a given region experience the same offshore ocean conditions. Our experiments thus emulate this approach by imposing the same "regional" background forcing in all of our simulations. In effect, we created an arbitrary ISMIP6 "region", then tested how much fjord conditions may vary within that region based on local forcing mechanisms (Lines 73-76).

Previous studies point to sill-driving mixing (and thus sill depth) as a primary mechanism for local water transformation and control on fjord water properties (e.g., Ebbesmeyer and Barnes, 1980; Cokelet and Stewart, 1985; Hager et al., 2022, Bao and Moffat, 2023), while fjord width, length, and depth are not expected to greatly influence water properties, just circulation (e.g., Carroll et al., 2018). Thus, we chose to focus on sill depth as the primary geometric constraint by using three different idealized geometries: S100, S250, and S400. These geometries were chosen

to span the depth range of the Atlantic-Polar Water thermocline, which is a ubiquitous feature around Greenland. As we draw our conclusions from the relative depths of the sill and thermocline (not absolute depths), we do not anticipate this choice will impact our results. We acknowledge this wasn't fully clear in the manuscript, so we have added a sentence explaining our choice to focus on sill depth as the dominant geometric constraint (Lines 85 – 88).

We have also added a sentence to the end of the conclusion stating "Limited observational evidence throughout Greenland supports the efficacy of these thermal forcing parameterizations, yet robust validation is still needed in realistic settings" (Lines 517– 518). This sentence is meant to to highlight the comparison of our results to observations in multiple locations in Greenland (Lines 363-372, lines 422-433, and Figure 5), but acknowledge some uncertainty exists when applying these parameterizations to realistic fjords.

**Specific comments:**

Abstract:

- Line 13: What the 2.9°C refer to is unclear, maybe rather give the maximum modification that the TD experiences.

  Sentence changed to: "Despite identical ocean boundary conditions, we find that the simulated fjord processes can modify grounding line thermal forcing by as much as 3°C, the magnitude of which is largely controlled by the relative depth of bathymetric sills to the Polar-Atlantic Water thermocline" (line 12 – 14)

- Line 15-17: It's unclear if your parameterisation includes bathymetry?

  Thank you making this point – we have added the word "additionally" to this sentence to make clear that the iceberg parameterization also includes the adjustment for bathymetry discussed in the previous sentence. (Line 16)

Introduction:

- line 31: Morlighem et al., 2019 is no projection, Jourdain et al., 2020, introduces parameterisations for Antarctica, so neither citation really fits to your sentence

  Both citations here are in reference to "simplifying parameterizations of oceanic boundary conditions" and not "sea level rise projections." To the authors' knowledge, Morlighem et al. (2019) is the original study that uses a thermal forcing parameterization that has a bathymetric adjustment, similar to ISMIP6melt. Jourdain et al. (2020) introduces ocean thermal forcing parameterizations used in the Antarctic ISMIP6 experiments. As this sentence is a general statement about parameterizations used in sea level rise projections, and is not specific to Greenland, we feel this citation is justified here.

  We acknowledge the original reading of this sentence was confusing in regards to the citations, so we have changed the sentence to read:

"To date, sea level rise projections have instead relied on poorly-validated simplified parameterizations of oceanic boundary conditions in ice sheet models – such as those developed in Morlighem et al. (2019), Jourdain et al. (2020), and Slater et al. (2020) – that are large sources of uncertainty when predicting future mean sea level (Goelzer et al., 2020; Seroussi et al., 2020)." (Lines 31 – 34)

- line 31: Seroussi et al. is for Antarctica, the citation does not fit here.
  This sentence is a general statement about the impact of thermal forcing parameterizations on the uncertainty of sea level rise projections and is not specific to Greenland. We therefore feel this citation is justified here.

- line 40: Smith et al., 2020 presents satellite observations of thickness changes, it does not link them to the ocean forcing, the citation does not seem to fit here.

  This citation has been removed from the text.

- Equation (1) here glacier front changes are directly linked to frontal melt changes, however, this misses out changes in ice dynamics: a glacier terminus could stay in the same position for higher melting when the ice discharge increases at the same time (at least for a while). This seems to be missing some physics?

  Yes, this is a crude parameterization of ocean-driven glacier retreat and is undoubtedly missing important physics, as is acknowledged in the paper describing the parameterization – Slater et al. (2019). Nonetheless, this is one of the two parameterizations for Greenland frontal ablation used by ISMIP6 experiments, and we only include it here only to provide context for the thermal forcing parameterization it uses. It is outside the scope of this paper to evaluate the legitimacy of this equation, as we are solely concerned with thermal forcing parameterizations.

  As described in response to the next comment, we have inserted a sentence at the end of the introduction stating our intent to focus on thermal forcing parameterizations, instead of assessing the validity of Eqs. 1 and 2.

- Explicitly state somewhere that you do not evaluate melt parameterizations, just the thermal forcing aspect. And state clearly, that the ISMIP parameterisations underlies a thermal forcing parameterisation, that the resulting melt is relevant, however, this is still open and here always done using the equation (2, except for the retreat parameterisation in ISMIP6).

  The purpose of this paper – to test the accuracy of the thermal forcing parameterizations – is stated in the title, abstract, Lines 70-72, Lines 76–79, and throughout. We include the ISMIP6 submarine melt parameterization here only for context and it is outside the scope of this paper to assess the validity of this equation. To avoid confusion, have added a sentence to the end of the introduction: "This paper focuses solely on the accuracy of thermal forcing parameterizations, while assessing the validity of Eqs. 1 and 2, which are provided here only for context, is left to other studies." (Lines 79 – 80)

- My understanding is that equation (2) is mainly used to put the importance of thermal driving differences in the context to melt rates and not suggested as a valid melt parameterisation? If this is correct, state it.

  Please see response to above comment.

Methods:

- line 121: Where was the background velocity implemented?

  This is a parameter within IcePlume and is needed to drive melt across the glacier face. The sentence was reworded to make it more evident where the background velocity was implemented (Lines 131 – 132).

- What about sea ice in MITgcm?

  Thank you for making this point, as this is one local forcing mechanism we ignore. To limit unconstrained parameters, we do not explicitly include sea ice in our experiments. However, the influence of sea ice melt is likely captured to some degree by the parameterization of surface iceberg melting through the IceBerg Package. As iceberg depths are prescribed using an inverse power law size frequency distribution, a vast majority of icebergs are very shallow and could be thought of as representing sea ice interspersed among larger icebergs. One caveat to this reasoning is that IceBerg does not account for latent heating and brine rejection caused by sea ice formation. While we do not expect this process influence near-glacier thermal forcing in deep fjords that flush frequently, it is possible sea ice could be an important factor in some shallow fjords. We have added a sentence to this effect at the end of Section 2.1: "We did not include sea ice formation in our simulations and do not expect the neglect of associated latent heating and brine rejection to significantly affect our results, particularly in deep fjords. However, it is possible sea ice formation could influence thermal forcing in some shallow fjords." (Lines 144 – 147).

- line 155: do you want a new paragraph for the sentence "We compare.."

  Thanks for the suggestion – this sentence has been reworded and incorporated into a new paragraph that includes other sentences moved up from the results (Lines 169 – 176).

- line 157: Does "modeled area-mean" mean that it is averaged over the entire depth? And above, is the TD at the grounding line the one from the lowest cell?

  Yes, the area-mean is an average across the entire area of the glacier face (both vertically and horizontally). The grounding line is located at the lowest cell of the glacier face. A definition of "grounding line" water properties was added to Lines 106 – 107. We then inserted clarification of the meaning of "area-mean" to Line 176.

- Table 1: Define better exactly how the thermal forcing is calculated (e.g., which grid cells are used, just the closest to the calving front or are they averaged? How is this handled with different resolutions?).

As with other "near-glacier" metrics, near-glacier thermal forcing is a 10-day average of the two rows of cells closest to the glacier face (as described in Lines 105-106). This is the same for both resolutions.

- in general, I miss more motivation for your methods, e.g., why do you want to quantify sill-driven mixing? Why do you use three thermal forcing metrics (and not just one)?

The focus on sill-driven mixing is motivated by previous studies (e.g., Ebbesmeyer and Barnes, 1980; Cokelet and Stewart, 1985; Hager et al., 2022; Bao and Moffat, 2023) that show fjord mixing is primarily restricted to sill regions and can influence near-glacier thermal forcing. A clarifying sentence was added to the beginning of Section 2.3 (Lines 186 – 187).

As discussed in Lines 382–399, there is currently a lack of consensus in the glaciology community about which thermal forcing metric is most relevant to tidewater glaciers. ISMIP6melt and ISMIP6retreat were each designed to represent different thermal forcing metrics, so when testing their efficacies, we had to ensure we were testing each parameterization in a manner that was consistent with its original definition. ISMIP6retreat was originally designed by Slater et al. (2020) to parameterize theta_z, while ISMIP6melt was designed by Morlighem et al. (2019) to parameterize theta_gl. (theta_z is the near-glacier thermal forcing averaged between 200-500m depth and theta_gl is the near-glacier grounding line thermal forcing; see Table 2). We therefore test ISMIP6retreat and ISMIP6melt by comparing to theta_z and theta_gl in our model, respectively, to ensure we are comparing equivalent quantities. theta_A was then later developed in this paper as an alternative to theta_z and theta_gl that is sensitive to other processes not captured by the original ISMIP6 parameterizations.

Lines 155 – 157 from the original text have been expanded and combined with original lines 276 – 288 to form two new paragraphs at the end of Section 2.2 (Lines 169 – 184 in the revised manuscript). We believe these changes will better explain our reasoning behind using multiple thermal forcing metrics to compare with the parameterizations.

- furthermore, I miss a motivation and explanation for the newly introduced melt parameterisations in the method.

The parameterizations introduced in the methods are the thermal forcing parameterizations used by Eqs. 1 and 2 to drive frontal ablation, and are not new melt parameterizations. We feel this is appropriately described throughout Section 2.2, particularly Lines 149-150 and Lines 166-168. We are open to feedback on how to clarify this distinction. Please also see our response to the reviewer's first general comment for more context on the order and use parameterizations in this paper.

Results:

- Line 210: Not sure where exactly you find the grounding line average salinity in the Figure? Is is simply the deepest value (at -800m)?

Grounding line water properties are taken from near-glacier cells at the base of the glacier, here at -800m. A definition of "grounding line" water properties was added at Lines 106 – 107.

- line 215: "..when iceberg keels extend… or subglacial discharge … below sill depth" – from the figure 2, this seems to be true for sill depth of -250 and -100m. How can you draw the logical conclusion that this is linked to the keel depth and vertical extend of the plume from this figure?

  In S400 runs, no icebergs extend below sill depth and water properties are entirely homogenous below sill depth. In S250 runs, only two runs have significant variability below sill depth; these are the two low resolution runs with iceberg keel depths extending to 400 m. Variability in these profiles only occurs in the upper 400 m and coincides with the input of iceberg melt water (and associated heat sinks) shown with Q_berg and H_berg in panels b and e. In all iceberg S100 runs, keels extend below sill depth and thus contribute to cooling of the entire water column (in combination with sill-driven reflux). Additional variability seems to coincide with the terminal plume depths shown in black and white triangles.

  Lines 215-217 of the original text (243-245 in revised text) states that variability below sill depth only occurs when iceberg keels extend below sill depth or when subglacial plumes reach neutral buoyancy below sill depth. This holds true for all runs, and water below sill depth remains homogenous in all runs where this is not the case.

- line 219-221: this is hard to see from Figure 2. At least in panel (e) it looks like there might be blue triangles left and right of black triangles (and the lines intersect above of -200m).

  Thanks for the suggestion - Blue and black triangles were removed from Figure 2. Instead, all thermal forcing metrics have been added to Table 1, which makes this information more apparent. Throughout Section 3.1, we then replace references to Figure 1 with references to Table 1 where appropriate.

- line 224: again, this refers to the middle and right columns, or how can this be seen more precisely in the figure?

  Please see changes to Figure 2 described in above comment

- line 237: the third EOF "depicts temperature variability coincident with the terminal depth of subglacial plumes" – I am not sure this is is very clear, e.g., the lower terminal plume depth around -400 m does not coincide with a change in the temperature profile? Why does this EOF not represent the reflux?

  The bottom cluster of terminal plume depths does coincide with a modulation in the shape of the third EOF mode, as do the approximate depths of the upper two clusters of terminal plume depths. In both this study and in Davison et al. 2022, reflux uniformly cools/warms the water column below sill depth (and should not alter water properties above sill depth), and thus we do not believe the third EOF mode could represent this process. While the authors acknowledge that the physical interpretation of EOFs modes can be ambiguous, we feel subglacial discharge is the most plausible explanation for third EOF mode. However, as we are least confident with this physical interpretation compared to the other EOF modes, we have changed line 237 of the original text to read: "A physical interpretation of the third EOF mode, contributing 5% of the variance, is less certain; however, this mode depicts temperature variability coincident with the terminal depths of subglacial discharge plumes (Figure 3b) that is not easily relatable to any

other forcing mechanism. We therefore interpret the third EOF mode to represent variable outflowing plume conditions." (Line 266 – 269 in the revised text).

- line 243-247: where are the absolute numbers? Can you add a table containing them?

  All freshwater fluxes have been added to Table C1 (now Table 1). References to this table were inserted to this sentence and succeeding sentences. (Lines 275, 278)

- line 248 – 250: this is simply because of the latent heat required to melt the icebergs, or?

  Yes, as is explained in Lines 331 – 335. We feel this is better served as a discussion point, because it is an interpretation of the data.

- Figure 2: Are the profiles from the center of the calving front or are they averaged over the calving face? I would mention earlier on that the columns are for the different sill depth, e.g., add this as titles to the columns. The figure is quite dense, you could help the reader by indicating what features they should look at in the figure. E.g. for the sentence in lines 212-214 "However, water properties…" you could add in the end ".. for S100 runs (compare the blue and black triangles indicating the depth-averaged thermal driving in the ocean simulations across the three lower panels). Same for lines 216, explain how the reader can see that "iceberg keels extend beyond sill depth" and "subglacial plumes reach neutral buoyancy". Same for the next sentence. It looks like some triangles are missing, e.g., there are no black triangles in panel (f)?

  As described in Lines 105-106, all "near-glacier" output is an average of all cells within two rows of the glacier face.

  Sill depth column titles were intentionally left out because the Figure 2 is already dense and it was the intent that the horizontal dashed line depicting sill depth could make this distinction. However, we have reorganized the Figure 2 caption to bring attention to the difference between figure columns earlier in the text.

  Black/blue triangles have been deleted from Figure 2 and the data incorporated into Table 1. The reference for Figure 1 at Line 242 is now for both Figure 1 and Table 1.

  Lines 215-217 of the original text was changed to "…with only minor variability occurring when iceberg keels extended below sill depth (see $Q_{berg}$ and $H_{berg}$ profiles in Figure 2) subglacial discharge plumes reached neutral buoyancy below sill depth (this most often occurs with line-plumes; see black/white triangles in Figure 2a-c)." (Line 244 – 246 of the revised text). Additionally, the last line of the caption will be changed to "The vertical distribution of iceberg freshwater fluxes (Qberg ) and heat fluxes (Hberg ) are provided in a-c and d-f, respectively, to depict the depth of iceberg melt relative to sill depth and profile variability."

- Figure 4b: What does this mean that there is higher reflux with higher freshwater input at depth?

  The greater the portion of freshwater input that enters the system below sill depth, the greater the portion of water that is refluxed at the entrance sill.

- equation 12: what is the motivation for this "skill score" definition, is this something commonly used?

  This is a commonly used metric to compare modeled profiles of ocean/atmospheric properties to observations and was originally developed in Willmott (1981). Citation to Willmott 1981 was added to text. This paragraph (and the equation therein) was moved to the Methods section (Lines 177 – 184).

Discussion:

- I would move part of the discussion to the results, e.g., the definition of the new parameterisations for thermal driving, how this is translated into melt.

  The main purpose of this paper is to evaluate the current ISMIP6 methods for parameterizing thermal forcing and identify the primary sources of error. The additional parameterizations introduced in Section 4.2 and the discussion on melt rate uncertainty (Section 4.3) should therefore be treated as an exploration of possible improvements to the current ISMIP6 methods and what impact this could have on ISMIP6 melt rates. Thus, we feel these sections are better suited as discussion points.

- Section 4.1: you are comparing unlike things here as you are using for 1. the average thermal driving as the relevant quantity, while in 2.-4. your relevant quantity is the variability in the thermal driving profile. If you want to list the processes "in order of importance", I suggest that you think about what defines their importance (relevant quantity is resulting basal melt rate, temperature profile or the average temperature) and then compare them with respect to this quantity.

  This is a good point and alludes to one of the main difficulties in establishing a thermal forcing parameterization that is useful for ice sheet models. As discussed in Lines 382-399, the best method for defining an ocean thermal forcing metric that is relevant to glacier frontal ablation processes is still a topic of ongoing debate. It is unclear if frontal ablation can be accurately determined solely from grounding line thermal forcing, mean thermal forcing, or an entire profile. At the same time, current submarine melt parameterizations cannot be corroborated by direct observations of melt at glacier termini. Therefore, there is no straightforward relevant quantity of interest for thermal forcing parameterizations. Instead, Section 4.1 determines the level of importance of each mechanism primarily based on its influence on full profile variability (including the translation of profiles that occurs between sill depth groups, which is equivalent to Theta_A), thus capturing any potentially relevant quantity of interest. When possible, we then use additional lines of evidence, such as heat fluxes, to further corroborate our ranking of important processes.

  Accordingly, the authors have made the following changes to the manuscript:

  1. Added to beginning of Section 4.1: "We assign importance to each local forcing mechanism based on its impact on near-glacier thermal forcing, which we corroborate with additional evidence where possible." (Lines 314 – 316).

2. Lines 169 – 176 have been changed in the manuscript to better explain why multiple thermal forcing metrics exist and the difficulty of establishing a relevant quantity of interest

3. Lines 441 – 446 were added to the beginning of Section 4.3 to explain why we do not compare our results to absolute submarine melt rates.

- lines 350 and following: is this caveat ("the dependence on specific depth when calculating thermal forcing") not the same caveat as discussed in the paragraph above, i.e., that sills are highly relevant for thermal forcing in the fjords?

  This sentence is referring to the ISMIP6 practice of defining Theta_z and Theta_gl at specific depths (Theta_z is only defined between 200-500 m and Theta_z is only defined at the grounding line). This sentence has been changed to "…namely that current parameterizations only define thermal forcing at specific depths (e.g., at the grounding line or between 200 – 500 m)." (Lines 374 – 375).

- give the equations for the parameterisations, e.g., how exactly follows the AMberg the Gade line (what ambient water masses do you assume to mix with, how much mixing occurs, see line 380)?

  The Gade slope equation has been added as Equation 13 (Line 407).

- line 386: "*iceberg* melting" instead of "submarine melting"?

  This has been changed in the new manuscript (line 411)

- Figure 5: ISMIP6retreat label should be AMretreat, or (this is also mixed up in the text)? Please add the other thermal driving parameterisations as well, i.e.,. AMmelt, AMconst as well as dots for the ISMIP6 ones. How well do they perform?

  This should be ISMIP6retreat, although the profiles used for ISMIP6retreat and AMretreat are the same.

  The profiles used to calculate all parameterizations have been added to Figure 5 along with their skill scores.

- line 405: ISMIP6melt is not on the figure 5, AMretreat shows higher temperatures. The difference could also stem from other reasons than "temporally varying conditions", i.e., horizontal variability in the sill and ice conditions...

- ISMIP6melt has been added to the plot, as with all other parameterizations. All observed profiles (gray) have warmer temperatures below sill depth than are predicted by the profile for AMberg (blue).

  We see no evidence of meaningful horizontal temperature gradients between the sill-driven mixing zone and glacier face in our modeling, regardless of iceberg distribution, subglacial discharge, etc. Observations of glacial fjords generally support this claim (e.g., Straneo et al., 2012; Mortensen et al., 2014, Moffat et al., 2018, Gladish et al., 2015). Therefore, the most

likely explanation is that the warm water observed at depth is a remnant of warmer water that had previously entered the fjord earlier in the summer. We have added the following sentence near the end Section 3.1 (Line 247 – 248): "Horizontal gradients in water properties were concentrated over the sill region but were negligible within the fjord (Movie S1)." We also added a supplementary movie, Movie S1, that shows 6 representative simulations from our study with weak horizontal gradients in each. MovieS1 is first introduced at Line 85.

Line 405 of the original text was then changed to: "We interpret this discrepancy to be the result of temporally varying conditions in Ilulissat Icefjord that reflect previously warmer conditions in Disko Bay, and not that of steady-state along-fjord temperature gradients, which were negligible in our runs (Movie S1)." (Lines 433 of revised text).

- Figure 6: Difference between each theta and what? The far-field theta/boundary conditions? How are sub-shelf melt rates calculated with (2) when using a thermal driving profile? Non-iceberg runs are black? I think also the green markers show the thermal forcing parameterisations differences in thermal driving relative to the boundary conditions / ISMIP6retreat case? Don't you model melt rates with MITgcm and IcePlume, why don't you compare to those as well?

  Figure 6 depicts the range of the various thermal forcing definitions (Theta_z, Theta_A, and Theta_gl) for each of our simulations. The green markers depict the range of all seven thermal parameterizations explored in the paper, grouped by sill depth and subglacial discharge. We replaced the phrase "Difference between" to "range of" in figure caption to make this clearer. A reference to Table 1, which now includes theta_z, theta_gl, and theta_A for all runs, was also added to Line 448. In addition, we inserted two new sentences to the Figure 6 caption that better explain how to interpret the figure:

  1. "ΔΘ equal to zero indicates perfect agreement between theta_z , theta_A, and theta_gl for a single run, while a large ΔΘ indicates these thermal forcing metrics vary substantially."
  2. "For green markers, ΔΘ equal to zero indicates perfect agreement between all seven thermal forcing parameterizations, while a large ΔΘ indicates substantial variation between parameterizations."

  We also changed the sentence "…all seven thermal forcing parameterizations *explored in this paper* for each sill depth and subglacial discharge group…" to "all seven thermal forcing parameterizations for each sill depth and subglacial discharge group…" to reduce wordiness.

  Thermal forcing parameterizations were computed as described in Sections 2.2 and 4.2, and the resultant scalar value is used as Theta in Eq. 2. The scalar values provided by each parameterization, not the profiles used to calculate the scalars, are used in Eq. 2 (line 166).

  The gray box in the legend was intended to be the same shade of gray as the markers, but there was a bug in the code. This has been fixed in the updated manuscript.

As discussed previously, we avoid comparing our results to absolute submarine melt rates because melt parameterizations are not yet reliable. Please see above description of changes made to the beginning of Section 4.3 (Lines 441 – 446).

- line 417: how much is the 200m/yr in relative terms (i.e., how large are the melt rates overall)?

This is a difficult comparison to make, as direct observations of tidewater glacier submarine melt rates do not exist in Greenland to our knowledge. However, projected melt rates (using Eq. 2) from Slater et al. (2020) indicate the choice of thermal forcing metrics and/or parameterizations could have a significant impact on projected submarine melt rates for some Greenland glaciers. To make this point, we have added a new sentence to Section 4.3: "Melt rate parameterization uncertainty of this magnitude (<= 1.7 m/d) could significantly impact long-term projections of submarine melt for some Greenland glaciers (compare to projected melt rates in Slater et al. (2020), their Figure 10)." (Line 453- 455).

- line 420: within a given run, theta_z, theta_A and theta_gl calculated by the forcing parameterisations differed…(or how did you estimate the difference)?

This sentence refers to the seven thermal forcing parameterizations explored in this paper referred to in the previous sentence (ISMIP6melt, ISMIP6retreat, AMberg, etc.), and not theta_z, theta_A, or theta_gl, which are metrics calculated from the model to test the parameterizations. The difference is the range of thermal forcing values given by the parameterizations.

To improve clarity, we changed "differed by" changed to "ranged from" (Line 451). Please also see our above changes to the Figure 6 caption regarding this point.

- line 434: Could it be that the reflux is hard to get from the EOF because it is linked to the bathymetry and you removed that in your EOF analysis?

That is likely the case. We had alluded to this in Section 3.1, but admit it was not fully clear. Lines 269-271 now read: "As reflux primarily affects the water column below sill depth, which is homogeneous and similar to depth-averaged water properties, variability resulting from reflux was likely removed with the depth-average when computing EOFs".

We have also changed Line 434 in the original text to: "As discussed in Section 3.1, the influence of reflux is difficult to discern with EOF analysis …" (Lines 467-468 of revised manuscript).

- line 484: "reduces error in *thermal driving profiles* compared to ismip6 estimates"..

Thank you for this suggestion – this sentence was change to:

"Although the updated parameterizations presented in this paper greatly reduces thermal forcing error compared to existing ISMIP6 methods…" (Line 481)

- line 461: add "in shallow silled fjords *in our idealised simuations*".

Thank you for this suggestion. This sentence has been changed to "… however, without accounting for bathymetry, parameterizations overpredict thermal forcing by at least 2∘C in our idealized shallow-silled simulations." (Lines 493 – 494)

- Add that it remains an open question which theta to usse, or how to translate the profile into melt rates.

An additional sentence was added to the text at Lines 502- without accounting for bathymetry, parameterizations overpredict thermal forcing by at least 502 – 503: "The uncertainty this contributes to thermal forcing in ice sheet models could be substantial, and determining where best to define thermal forcing at a calving face should be a topic of high importance."

- I miss a discussion of the next steps (develop and evaluate melt parameterizations) and caveats (idealized model domain, only one background forcing, comparing to one fjord,...).

Multiple next steps are discussed in the conclusion. These include (1) development of an iceberg prevalence prediction method so that AMfit can be used accurately (Lines 509 - 510), and (2) the development of a box-model parameterization that includes reflux (Lines 511 - 515). As noted above, we will also add a sentence highlighting the importance of determining which thermal forcing metric to use in melt parameterizations. The development and evaluation of melt/retreat parameterizations is a related but separate area of research. Please also see the preceding response regarding the sentence added at Line 502 – 503.

Lines 363-372, lines 422-433, and Figure 5 are aimed at broadening the results of our idealized simulations to observations of multiple Greenland fjords. We have also added a sentence to the conclusion: "Limited observational evidence throughout Greenland supports the efficacy of these thermal forcing parameterizations, yet robust validation is still needed in realistic settings." (Lines 517 – 518)

The choice to use one background temperature/salinity forcing was made to be consistent with the design of ISMIP6 experiments, and temperature/salinity profiles at the open boundaries were chosen to represent conditions surrounding Greenland. The feature of importance in the boundary conditions is the depth of the Atlantic-Polar water thermocline in relation to the sill depth. While the depth of this thermocline may change throughout Greenland, we base our findings only on its depth relative to sill depth, and thus, we do not anticipate this choice impacting our results.

Appendix:

- State what the abbreviation TEF stands for.

TEF is now defined in the Appendix A header in addition to its original definition in Line 189.

- line 489: what do you mean with volume conservation? In general, mass, energy and momentum are conserved, but volume might change with density (temperature, salinity, pressure) changes. Please explain.

The TEF framework is based on the assumption that the mass and volume entering the mixing zone must equal the mass and volume exiting the mixing zone.

- Equation A4: How do these follow from "mass and volume conservation"? Is it rather that you assume salinity on the glacier side is lower because of mixing with melt water?

  Yes, TEF is specifically designed for estuaries where there is a freshwater source at the head of the estuary. Therefore, the freshest layer must be the outward flowing (upper) layer on the glacierward side of the mixing zone, and the densest layer must be the inward flowing (lower) layer on the ocean side of the mixing zone. Then due to mixing across the sill, the remaining layers must have a density somewhere between these two. This is described in detail within the sources cited in this section.

- Figure C1: This is the algorithm for AMfit, right? The definition of effective depth could be repeated here, or you could point to the relevant location in the methods.

  This is the step-by-step process for computing AMberg and AMmelt, which when implemented in iceberg and non-iceberg laden fjords, respectively, make AMfit. "(Section 2.2)" was added to "Define Effective Depth" bubble

- Table C1: Please add theta_gl and theta_z here as well.

  Thanks for the suggestion - Theta_gl and Theta_z have been added to Table 1 (formerly Table C1)

---

## Author Response (AR3)

Dear Alexander Hager and co-authors,
I am happy to recommend your manuscript for publication in The Cryosphere pending a few technical corrections which I outline below.
Best,
Nanna B. Karlsson, TC Co-editor-in-chief

> Thank you for your handling of the manuscript, we are happy to hear you are recommending it for review. We have addressed your technical corrections, which are outlined below.
>
> Best,
> Alex

Additional private note (visible to authors and reviewers only):
Technical corrections

In the manuscript and figure captions, the terms line-plume and sheet plume are used. For example, Table 1 uses line-plume, Fig. 2 uses sheet plume in the figure legend but line-plume in the caption. Fig. 6 uses sheet plume in the caption. If the two terms refer to the same phenomenon, please use only one of the terms. If there is a difference between the two, please clarify this.

> Line-plume are sheet plume are used synonymously and we agree this is confusing. We have changed all references to a "sheet plume" to "line plume" (lines 257, 299, Figure 6 caption, Figure 2 legend, Figure 4 legend). To be consistent, we have also changed any "line-plume" reference to "line plume" (Lines 127, 129, 246, and 290, as well as Table 1 caption, Figure 2 caption, and the Figure 4 caption).

Fig. 4, caption re. (a): "Horizontal blue and orange rectangles depict ranges of estimated surface heat fluxes..." I see faint orange and blue horizontal rectangles in (a) but the most prominent is a purple rectangle which is not mentioned. Is this a typo?

> The purple rectangle was intentionally created from a translucent blue rectangle overlapping a translucent orange rectangle. To fix this, we have removed the translucent fill of these two rectangles and instead use the rectangle outlines.
>
> p. 22, 445-446: Typo, "... which was to prescribed submarine..." -> "... which was to prescribe submarine..."

> This typo was fixed.

A note on Greenlandic place names:
The official name of the fjord that Kangiliup Sermia discharges into is Kangilleq (p. 5, line 135-136 and other places). If you would rather avoid introducing this name as you only need it a few times, you could rephrase the sentences where needed.
The official name of Ilulissat Icefjord is Kangiata Sullua. You could mention the official name when introducing Ilulissat Icefjord and then continue using Ilulissat Icefjord for clarity.

"Kangilliup Sermia fjord" was changed to "Kangilleq" on Lines 137 and 142.

"Ilullissat Icefjord" was changed to "Kangiata Sermia" on Lines 17-18, 124, 143, 372, 424, 428, 432, 433, 458, 481, 483-484, 499, 522, and 542, as well as the Figure 5 caption/legend.